# Forecasting Future World Events
# with Neural Networks

**Andy Zou**
UC Berkeley

**Tristan Xiao**
UC Berkeley

**Ryan Jia**
UC Berkeley

**Joe Kwon**
MIT

**Mantas Mazeika**
UIUC

**Richard Li**
UC Berkeley

**Dawn Song**
UC Berkeley

**Jacob Steinhardt**
UC Berkeley

**Owain Evans**
University of Oxford

**Dan Hendrycks**
UC Berkeley

## Abstract

Forecasting future world events is a challenging but valuable task. Forecasts of climate, geopolitical conflict, pandemics and economic indicators help shape policy and decision making. In these domains, the judgment of expert humans contributes to the best forecasts. Given advances in language modeling, can these forecasts be automated? To this end, we introduce Autocast, a dataset containing thousands of forecasting questions and an accompanying news corpus. Questions are taken from forecasting tournaments, ensuring high quality, real-world importance, and diversity. The news corpus is organized by date, allowing us to precisely simulate the conditions under which humans made past forecasts (avoiding leakage from the future). Motivated by the difficulty of forecasting numbers across orders of magnitude (e.g. global cases of COVID-19 in 2022), we also curate IntervalQA, a dataset of numerical questions and metrics for calibration. We test language models on our forecasting task and find that performance is far below a human expert baseline. However, performance improves with increased model size and incorporation of relevant information from the news corpus. In sum, Autocast poses a novel challenge for large language models and improved performance could bring large practical benefits.

## 1 Introduction

Forecasting plays a crucial role in the modern world. Climate forecasts shape the policies of governments and companies (Gillingham et al., 2018). Economic forecasts influence investment and employment (Christensen et al., 2018). In 2020, forecasts about the spread of COVID-19 led to national lockdowns and border closures (Adam, 2020), slowing the spread of the virus. Consequently, machine learning (ML) models that make accurate forecasts across a broad range of topics could enable more informed decision making at scale and improve ML safety (Hendrycks et al., 2021c).

Two main approaches to forecasting are described in the forecasting literature: statistical and judgmental forecasting (Webby and O'Connor, 1996; Armstrong, 2001). In *statistical forecasting*, forecasts are made by traditional statistical models for time-series prediction such as autoregression (Makridakis et al., 2008) or by ML time-series models (Makridakis et al., 2020; Triebe et al., 2021). Humans create and tune the models but do not tweak individual forecasts. This works well when there are many past observations of the variable being forecast and minimal distribution shift. By contrast, in *judgmental forecasting* human forecasters use their own judgment to determine forecasts. The forecasters may use statistical models, but often integrate information from various sources including news, accumulated knowledge, and *a priori* reasoning. This enables forecasting for questions where

36th Conference on Neural Information Processing Systems (NeurIPS 2022) Track on Datasets and Benchmarks.

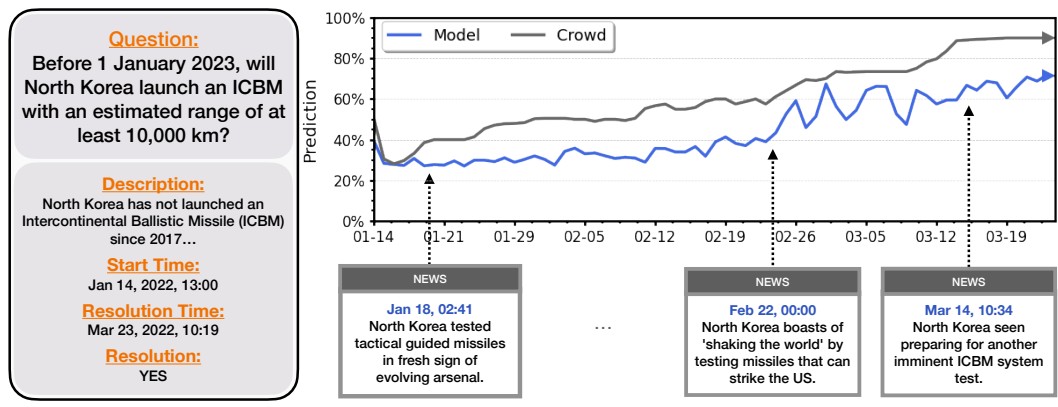

Figure 1: Example from the Autocast dataset, including the question, the resolution of the question, and the timeseries of aggregate human expert forecasts (Crowd) from the start date to the time the question resolves. We train a language model to generate forecasts at each timestep, using only news articles available at that timestep (i.e. without allowing any leakage of information from the future).

past data is scarce or subject to distribution shift (Tetlock and Gardner, 2016). For brevity, we refer to judgmental forecasting as "forecasting" in the rest of the paper.

Because it relies on scarce human expertise, forecasting is only used for a small number of questions. This motivates using ML to automate forecasting, e.g. by automating human information retrieval (finding news sources), reasoning (to decide if some evidence bears on a forecast), and quantitative modeling. ML models may also have some advantages over human forecasters. Models can read through text or data much faster than humans and can discern patterns in noisy high-dimensional data that elude humans. When it comes to learning, humans cannot be trained on past data in manner simulating actual forecasting (e.g. How likely was the Soviet Union's collapse from the viewpoint of 1980?) because they know the outcomes – but past data can be used for ML models.

As a step towards automating human forecasting, we introduce *Autocast*, a new dataset for measuring ML models' forecasting ability. Autocast includes thousands of forecasting questions collected from human forecasting tournaments. The questions vary in the forecasting horizon from days to decades, in the topic (including politics, economics and science), and in the answer format (e.g. multiple-choice vs. predicting a number). The questions are pre-selected for public interest, and there is a strong human baseline (the crowd aggregate of many competitive forecasters). The questions in Autocast are about past events (e.g. the US 2020 election) and so ML models could answer them simply by memorizing what happened. To test forecasting ability, we need to simulate the state of information *before* the past events ("retrodiction"). To this end, we curate a corpus of news items from Common Crawl (Nagel, 2016) that is organized by date. This means a model can be exposed only to news from before the outcomes being forecast, allowing for a rigorous test of retrodiction.

We implement a number of baseline models on Autocast, and demonstrate how language models can be trained on past forecasting questions by retrieving from our news corpus. We find that performance improves with model size and that information retrieval helps. However, all baselines are substantially worse than aggregate human forecasts. On forecasting binary outcomes, the best ML model achieves 65% accuracy vs. 92% for humans (and 50% for random). The same ML model (Raffel et al., 2020) is close to the human ceiling when fine-tuned on other NLP benchmarks (e.g. SQuAD from Rajpurkar et al. (2016)), which shows that Autocast is a challenging, real-world test for ML. Experiment code and the dataset are available at github.com/andyzoujm/autocast.

**Contributions.**

1. We introduce Autocast, a dataset for forecasting that covers diverse topics (e.g. politics, economics, society, science) and varying time horizons.

| Question Summary | Category | Answer Type | Resolution |
|---|---|---|---|
| Will a Tesla car demonstrate fully autonomous capability before the end of 2021? | Science & Tech | T/F | No |
| What will be Putin's approval rating value 3 months after the potential invasion of Ukraine? | Politics | Numerical | 83 |
| When will the US-Canada border reopen? | Social | Numerical | Nov 8, 2021 |
| How many vacancies will arise on the U.S. Supreme Court in 2021? (A) 0 (B) 1 (C) 2 (D) 3 or more | Economy | MCQ | A |

Table 1: Examples from the Autocast dataset. For brevity, we do not depict the full question specification, which often includes context, definitions, and detailed resolution criteria.

2. Part of our dataset is a large news corpus organized by date, allowing us to rigorously evaluate model performance on historical forecasts.

3. We show that forecasting is challenging for current language models, with accuracy and calibration far below a strong human baseline.

## 2 Related Work

**Forecasting.** A recent experiment (Kirk Bonde, 2022) tested GPT-3 in the few-shot setting on true/false questions collected from Metaculus (one of the sources for Autocast). However, since questions were not filtered by date, some answers would have appeared in GPT-3's training data. Similar to our work, ForecastQA (Jin et al., 2021) is a dataset of forecasting questions that covers a range of topics. However, ForecastQA's questions were written by crowdworkers without forecasting experience. Consequently, the questions are often nonsensical or ambiguous given the lack of additional context, e.g. "To how many people will the Representative of an internet speak to by September 2019?", or "In July 2019, will an article say there were no volunteers in 2016?". We found that a high percentage of ForecastQA questions suffer from these issues. By contrast, our questions were written by experienced forecasters and are always unambiguous given the full question description. Finally, ForecastQA's human baseline was done retrospectively (making it unrealistic) whereas our dataset contains expert human forecasts from real forecasting questions.

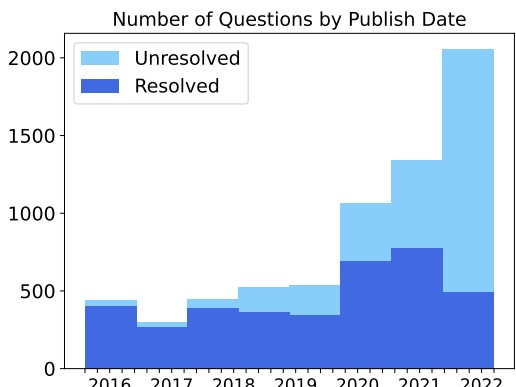

Figure 2: The number of questions in Autocast by publish date. Unresolved questions are about events after 2022 (e.g. the 2024 US Election). They are not included in the test set but can be used as auxiliary training data. Note that the number of questions is accelerating. Future questions will be added to Autocast, improving it over time.

**Information Retrieval.** Information retrieval is crucial for forecasting, as good forecasts depend on up-to-date, specialized information drawn from multiple sources (Tetlock and Gardner, 2016). Recent work has used information retrieval to improve question-answering in large language models (Lewis et al., 2020; Nakano et al., 2021; Shuster et al., 2021) or to address time-sensitive questions (Chen et al., 2021). This has been applied to tasks that are related to forecasting, such as fact checking and truthful question-answering. In forecasting, it is useful to read and compare multiple news articles daily, in order to build an accurate picture of the current state, and then to iterate this process. We

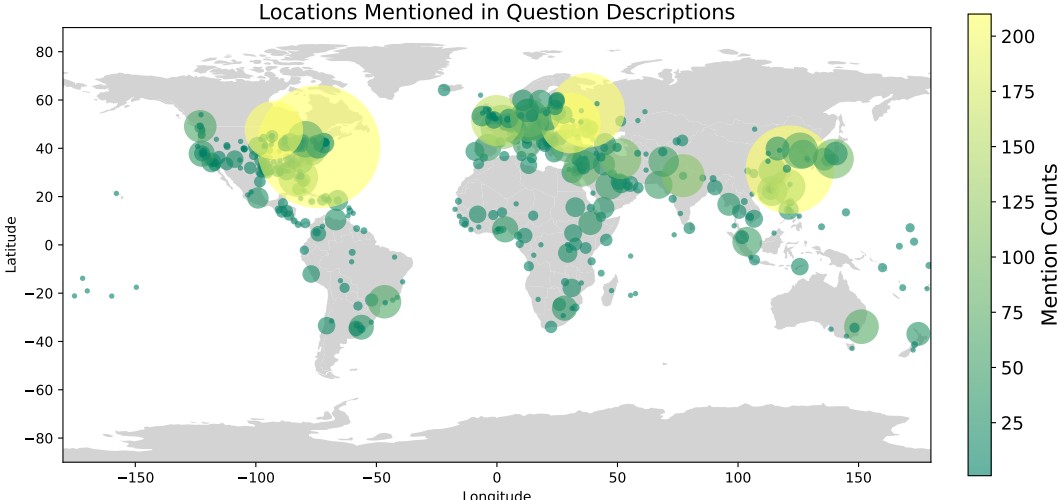

Figure 3: Autocast contains questions about locations across the world. The questions in the dataset mention over 500 cities, spanning six continents.

design an architecture for this purpose (albeit with limits on article length and time horizon), drawing inspiration from Wang and McAllester (2020).

**Calibration.** Calibration is important in forecasting (Tetlock and Gardner, 2016). Even expert forecasters will be highly uncertain about some outcomes of interest. Such forecasts will be more useful in the form of calibrated probabilities than as point estimates. Thus forecasters are evaluated with proper scoring rules, which incentivize calibration. There is an extensive literature on improving the calibration of deep learning models (Guo et al., 2017; Nguyen and O'Connor, 2015; Lin et al., 2022; Minderer et al., 2021; Kull et al., 2019), mostly for classification with a fixed set of classes. One part of Autocast requires models to forecast continuous quantities varying over multiple orders of magnitude, which has not been explored in prior work.

**Truthful question-answering.** Current language models often generate falsehoods when answering questions (Shuster et al., 2021; Lin et al., 2021), and they also achieve poor calibration when giving probabilistic answers (Hendrycks et al., 2021a) to human knowledge questions. However, for questions with a known ground truth answer, we expect models to improve as a result of scale, fine-tuning, and information-retrieval from reliable sources (Bai et al., 2022; Nakano et al., 2021; Hadfield-Menell et al., 2016; Turner et al., 2020; Wainwright and Eckersley, 2019). Yet humans also want models to give calibrated and truthful answers to questions that are too difficult or costly for us to answer ourselves (Irving et al., 2018; Evans et al., 2021; Leike et al., 2017; Hendrycks et al., 2021d; Reddy et al., 2020; Nahian et al., 2021). Forecasting is useful for this purpose. Forecasting questions are challenging but eventually become easy to evaluate. By contrast, it may be difficult for humans to evaluate superior answers to open problems in fundamental philosophy or science.

## 3   The Autocast Dataset

**Forecasting Questions.** We collected all available forecasting questions from three public forecasting tournaments (Metaculus, Good Judgment Open, and CSET Foretell), which resulted in 6,707 questions total. These questions tend to have broad public interest (e.g., national rather than local elections) and clear resolution criteria. Most questions are not already covered well by specialized forecasts (such as weather forecasts). The questions are either true/false, multiple-choice, or involve forecasting a numerical quantity or date (see Table 1 for examples). In these forecasting tournaments, participants begin forecasting a question on a given day (the "start date") and update their forecasts multiple times up until the "close date." At some later time, the forecast is *resolved* and participants are scored based on all their forecasts. (Note the resolution date is often just after the closing date but not always. The resolution can also happen *before* the planned closing date: e.g. when forecasting when an event will occur.) Thus the "crowd" forecast (which aggregates over participants) is a time-series of forecasts from the start to close date.

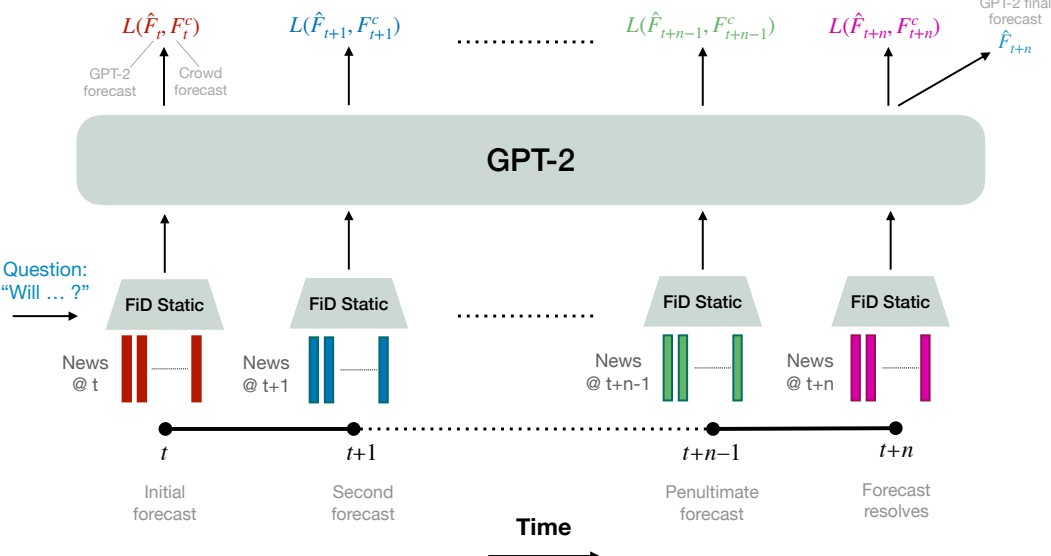

Figure 4: **Illustration of our FiD Temporal model.** Forecasts are made each day (from start date to resolution) by GPT-2. The input to GPT-2 is the top-1 daily news article retrieved by BM25, which is encoded by FiD Static (a T5 model). In training, GPT-2's target is the average of daily crowd predictions (denoted '$F_t^c$' for day $t$) and the resolved outcome. Like human forecasters, GPT-2 accumulates news information over time and updates its predictions.

Autocast includes the question, the start and close dates, the answer (if the question has resolved), and the time-series of crowd forecasts (Figure 1). Half of the questions have not yet resolved and correspond to ongoing tournaments. Some of these questions concern events decades in the future, requiring reasoning over long time horizons. These questions can still be used as training data by using the crowd forecast as the target (as a high-quality proxy for the ground truth). However, the test set only includes resolved questions. Our dataset also includes metadata that is helpful for forecasting. There is detailed background information about the question (including precise terms of resolution) and also links to relevant information posted by tournament participants. We include more details in the appendix.

**Train and test split.** It is standard in ML for the test set to be drawn from the same distribution as the train set. However, randomly splitting our questions into train and test without considering the date would not simulate the conditions of forecasting. For example, a test question ("Will Trump win the 2020 election?") could come from an earlier date than a related training question ("Will President Biden pass the stimulus?"). Thus, we split our questions using a date cut-off of mid-2021, which means that questions in the test set resolve from mid-2021 to mid-2022. Note that if a model is pre-trained on data from after mid-2021, this will also not simulate forecasting faithfully. In both train and test sets, we implement dataset balancing for the true/false questions. To flip a label, we negate the question using OpenAI's GPT-3-175B Edit model (Brown et al., 2020) and manually check for correct negation.

**Contemporaneous news as context for forecasts.** When a human is making a forecast at time $t$, they use past and present ($\leq t$) information sources but are not exposed to any information from the future ($> t$). If they forecast again at $t + 1$, they will have updated on new information that was generated from $t$ to $t + 1$. These conditions can be simulated for ML models by (a) pre-training on text generated before time $t$, and (b) providing the model with new information generated between $t$ and $t + 1$. To this end, we provide a corpus of news articles scraped from CommonCrawl news (Nagel, 2016; Hamborg et al., 2017) that is organized by publish date. The articles were derived from diverse sources between 2016 to mid-2022 and total more than 200GB of data.

| Model | Parameters | T/F | MCQ | Numerical | Score | Average |
|---|---|---|---|---|---|---|
| Random | – | 50.0 | 22.1 | 34.5 | 18.8 | 18.8 |
| UnifiedQA | 0.2B | 45.4 | 23.5 | 34.5 | 17.2 | |
| | 0.8B | 48.2 | 23.5 | 34.5 | 18.6 | 19.5 |
| | 2.8B | 54.9 | 25.1 | 34.5 | 22.8 | |
| T5 | 0.2B | 61.3 | 24.0 | 20.5 | 32.4 | |
| | 0.8B | 60.0 | 29.1 | 21.7 | 33.7 | 32.9 |
| | 2.8B | 60.0 | 26.8 | 21.9 | 32.5 | |
| FiD Static | 0.2B | 62.0 | 29.6 | 24.5 | 33.5 | |
| | 0.8B | 64.1 | 32.4 | 21.8 | 37.4 | 37.2 |
| | 2.8B | **65.4** | 35.8 | 19.9 | **40.6** | |
| FiD Temporal | 0.6B | 62.0 | 33.5 | 23.9 | 35.8 | |
| | 1.5B | 63.8 | 32.4 | 21.0 | 37.6 | **37.8** |
| | 4.3B | 62.9 | **36.9** | **19.5** | 40.1 | |
| Human Crowd | – | 92.4 | 81.0 | 8.5 | 82.5 | 82.5 |

Table 2: Model accuracy on the Autocast dataset for each question type: true/false (T/F), multiple-choice question (MCQ), and numerical (Numerical). For Numerical, lower is better. For other metrics, higher is better. The model FiD Static (based on T5) retrieves the top 10 news articles over the period, while FiD Temporal (based on GPT-2 with T5 encoder) retrieves the top 1 article each day. Averaging over all model sizes, we find that the FiD Temporal achieves the best average.

## 3.1 Dataset Analysis

**Distribution of Questions.**  The questions in Autocast cover a very wide variety of topics. We divide the questions into five main categories: Economy, Politics, Science, Social, and Other. Each category contains numerous subcategories for a total of 44 subcategories ranging from foreign policy to AI. We list all subcategories in the Supplementary Material. The questions also cover a wide geographical distribution, as shown in Figure 3. Overall, Autocast tests both breadth of subject matter and depth (since questions ask for quantitative predictions about a specific, operationalized variable).

**Adding new questions over time.**  The number of questions submitted to forecasting platforms is rapidly increasing (Figure 2). If trends continue, in two years there will be twice as many questions available. Autocast is a living dataset and will be updated periodically with new questions. This will provide both more data for training and a new set of test questions (to assess overfitting).

**Human forecasts.**  The human crowd forecast for a given question becomes more accurate from the start to closing date, as shown in Figure 6. This is what we would expect if humans are updating their forecasts as more information comes out. In contrast to most ML benchmarks, the human crowd judgments are probabilistic. This allows us to evaluate their calibration. In the Supplementary Material, we show that crowd forecasts are well-calibrated.

**Distribution shift.**  We expect a distribution shift over time in both the questions being asked and in the answers. For example, there will be fewer questions about Ukraine before 2022. This distribution shift is inherent to forecasting and so it is crucial that models can manage it. We do find a shift in the distribution of question categories. For example, the number of questions in the Social category increased from 12.6% in the training set to 28.2% in the test set, possibly due the Covid-19 pandemic (which is included in Social).

## 4 Experiments

### 4.1 Baselines

The *Crowd* baseline uses the final aggregate human forecast before the closing date. The *Random* baseline uses the analytically computed random accuracy for true/false and multiple-choice questions.

For numerical questions, random predictions are sampled uniformly from the bounded range of possible answers specified in each question.

**Models without retrieval.** We evaluate *UnifiedQA-v2* (Khashabi et al., 2022) and *T5* (Raffel et al., 2020) models of various sizes. These models are trained on a variety of tasks, enabling strong generalization on many unseen language tasks. Using zero-shot prompting for UnifiedQA, we report results on classification questions. The UnifiedQA models were not trained on numerical questions, hence, we report random performance to enable comparison with other baselines. T5 is fine-tuned using its original output head for true/false and multiple-choice questions. To output numerical answers with T5, we add an additional linear output head.

**Retrieval-Based Methods.** We investigate whether retrieval models can improve performance by selecting relevant articles from the news corpus included with Autocast. Importantly, news articles after the close time or resolution time of a question are not available for retrieval, so retrieved articles only include information about the past. For all retrieval methods, we use Fusion-in-Decoder or FiD (Izacard and Grave, 2021) to encode articles retrieved by BM25 (Robertson et al., 1994; Thakur et al., 2021) with cross-encoder reranking. FiD uses T5 to encode retrieved passages along with the question and can be viewed as a minimal extension of T5 for incorporating retrieval. We truncate retrieved articles to a maximum length of 512 tokens.

The *FiD Static* baseline uses the top 10 retrieved articles after reranking, which is the standard method for retrieval-augmented prediction. The *FiD Temporal* baseline leverages the intermediate crowd predictions (before the question is resolved) as auxiliary supervision. The intuition is that crowd predictions will change based on

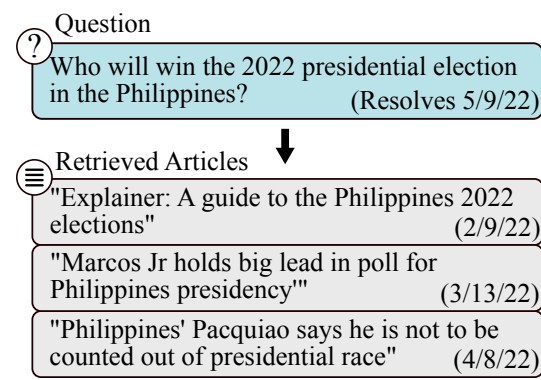

Figure 5: Articles retrieved by BM25 for a Politics question in the Autocast dataset with publication dates in parentheses. The articles are retrieved from 200GB of news and are highly relevant to making an informed forecast.

rational incorporation of new evidence, and these updates will not be captured by just training on the final outcome. For each day between the question's open and close date, we generate an embedding of the top news article using the frozen fine-tuned FiD Static model. These embeddings are then treated as input embeddings to an autoregressive model (GPT-2 (Radford et al., 2019)), which is fine-tuned to predict the average of the daily crowd prediction and the ground truth outcome. We illustrate this method in Figure 4. Figure 1 shows predictions from an FiD Temporal model over time for an example question.

## 4.2 Metrics

For true/false and multiple-choice questions, we evaluate models using percent accuracy. For numerical questions, we use $\ell_1$ distance, bounded between $0\%$ and $100\%$. We denote these question types as T/F, MCQ, and Numerical, respectively. To evaluate aggregate performance, we use a combined Score metric $(\text{T/F} + \text{MCQ} - \text{Numerical})/2$, which has an upper bound of $100\%$. A score of $100\%$ indicates perfect prediction on all three question types. Note that since numerical question responses are normalized between $0\%$ and $100\%$, the combined Score metric is lower-bounded at $-50\%$. We also report the Average score, which averages the combined metric of all model sizes.

## 4.3 Forecasting Evaluation

**Setup.** We fine-tune the T5 baseline for 10 epochs with a batch size of 8, an initial learning rate of $5 \times 10^{-5}$ with linear decay schedule, and a weight decay of $1 \times 10^{-2}$. The maximum sequence length of the T5 model is set to 512. We train FiD Temporal models for 5 epochs with a constant learning rate of $5 \times 10^{-5}$. Hyperparameters are selected based on early experiments. Additional details are in the Supplementary Material.

**Results.** We show results in Table 2. Although UnifiedQA-v2 obtains strong performance on various natural language benchmarks, it obtains close to random zero-shot performance on Autocast, showing the difficulty of forecasting. Fine-tuned T5 performs better, but multiple-choice accuracy is still at nearly random chance levels. Retrieval-based methods substantially outperform both UnifiedQA-v2 and T5, showing a relative increase in the Average score of 93% and 15%, respectively. Moreover, retrieval-based methods become more effective as parameter count increases, which suggests that the models learn to extract relevant information from retrieved articles.

Comparing the FiD Static and FiD Temporal baselines, we see that the Average score is slightly higher for FiD Temporal. However, the largest FiD Static model has the highest individual score. Thus, our temporal training strategy for incorporating the auxiliary crowd predictions neither harms nor helps compared to the static retrieval baseline. Future work could develop more effective ways of using these auxiliary training signals.

### 4.4 Model Analysis

**Relevance of Retrieved Articles.** We find that the retrieved articles are often highly relevant to the question. In Figure 5, we show examples of articles retrieved by BM25 from the news corpus in Autocast. Baseline models have access to the article text, but for brevity we only show the article title. The articles give information that is clearly relevant for making an informed forecast. Note that the T5 backbone for the baselines was pre-trained on data from before 2020, far before the timeline of the question, so retrieval provides vital information that models would not otherwise have. This suggests that large improvements on Autocast could come from integrating information from retrieved articles more effectively. We expect that more sophisticated retrieval methods would also improve performance, although efficiency becomes a concern when using large retrieval methods.

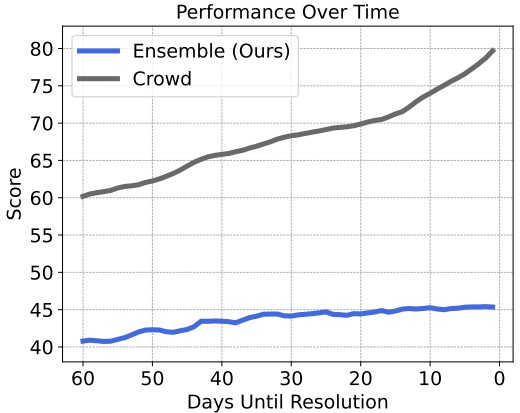

Figure 6: For the crowd and an ensemble of the two largest FiD Temporal models, prediction score increases as the resolution date grows nearer. This trend may be due to more relevant information becoming available over time, which the model can access through retrieval from the news corpus.

**Detailed Performance.** In the Supplementary Material, we show the performance of baseline methods on a more granular level. The per-category results indicate that Science & Technology is the most challenging category for models, whereas human forecasters have relatively consistent performance across categories. Inspecting the subset of questions that have been active for at least two months, we also find that the accuracy of the human crowd forecast and a model ensemble steadily increases over time up to the resolution date (Figure 6). This is to be expected, as more information about the eventual outcome is available closer to the time (e.g. election polls become more accurate). For this plot, we show an ensemble of the two largest FiD Temporal models, which has slightly higher final performance than the individual models and a clearer trend over time.

## 5 Calibrated Prediction of Numerical Quantities

In our results, we evaluate baselines on the accuracy of their point estimates, rather than their calibration. However, the eventual goal for Autocast is for models to achieve good calibration as well as accuracy. Here we describe an auxiliary dataset that helps with this goal for the challenging case of calibration on numerical questions.

**The IntervalQA Dataset.** In the Autocast training set, numerical quantities range over many orders of magnitude. Furthermore, Autocast has fewer than 1,000 numerical training questions. This problem of making *calibrated* predictions for quantities over many orders of magnitude using text inputs has not been addressed in work on calibration for language models. To this end, we curate

| Parameters | Point Estimate Distance | Conf. Interval Length | RMS Calibration Error |
|---|---|---|---|
| 22M | 20.8 | 2072.4 | 19.1 |
| 44M | 20.3 | 1115.7 | 16.6 |
| 86M | 19.6 | 763.1 | 16.9 |
| 304M | **18.1** | **305.4** | **13.5** |

Table 3: Results for DeBERTa-v3 models trained to output confidence intervals on our dataset of numerical predictions. The high dynamic range of the targets leads to large confidence intervals, but median interval size decreases with larger models as does RMS Calibration Error.

IntervalQA, an auxiliary dataset of numerical estimation problems and provide metrics to measure calibration. The problems in the dataset are not forecasting problems but instead involve giving calibrated predictions for fixed numerical quantities. The questions were sourced from NLP datasets covering diverse topics and with answers varying across orders of magnitude: SQuAD, 80K Hours Calibration (80k, 2013), Eighth Grade Arithmetic (Cobbe et al., 2021), TriviaQA (Joshi et al., 2017), Jeopardy, MATH (Hendrycks et al., 2021b), and MMLU (Hendrycks et al., 2021a). We filtered these datasets for questions with numerical answers, which yielded about 30,000 questions.

## 5.1 Metrics

We evaluate whether confidence intervals are calibrated. Concretely, if a method outputs $80\%$ confidence intervals on each test example, we would like the true prediction target to fall inside of these intervals $80\%$ of the time. Additionally, we would like for models to be calibrated across their entire dynamic range of outputs. To measure this, we compute *RMS Calibration Error* similarly to Nguyen and O'Connor (2015) and Hendrycks et al. (2019), but with fixed confidence levels $c \in \{50\%, 55\%, \ldots, 95\%\}$ and such that calibration is sensitive to dynamic range. We describe this metric in detail in the Supplementary Material. Low RMS Calibration Error indicates that models are calibrated across their entire dynamic range. We also compute the median prediction error between the predicted point estimate and the ground-truth target (*Point Estimate Distance*) and the median interval length averaged across all confidence levels (*Conf. Interval Length*).

## 5.2 Experiments

We fine-tune DeBERTa-v3 models (He et al., 2020) to predict a point estimate and a set of confidence intervals corresponding to the confidence levels in the RMS calibration error metric. On a high level, we use a loss with three components: (1) MSE loss between the predicted point estimate and the ground-truth target, (2) MSE loss between the boundaries of the predicted confidence intervals and the ground-truth target for boundaries that are on the wrong side of the target, (3) a penalty on the length of the predicted intervals to encourage finer predictions. The models are trained for 5 epochs with a batch size of 100. A detailed description is in the Supplementary Material. We show results in Table 3. All three metrics decrease with model size.

## 6 Conclusion

We introduced Autocast, a dataset for measuring the ability of neural networks to forecast future world events. The dataset contains thousands of forecasting questions from public forecasting tournaments, including ground truth outcomes and aggregated human predictions. We also curated a large corpus of news items from the Common Crawl news corpus, enabling rigorous evaluations without information leakage. We evaluated numerous baseline algorithms and demonstrated that model size and information retrieval can improve forecasting performance. To better evaluate calibration for numerical prediction, we introduced IntervalQA, a large collection of numerical prediction questions with a wide dynamic range of prediction targets, and evaluated state-of-the-art language models. Our results show significant room for future improvement.

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
