# Forecasting Future World Events
# with Neural Networks
# Supplementary Material

**Andy Zou**
UC Berkeley

**Tristan Xiao**
UC Berkeley

**Ryan Jia**
UC Berkeley

**Joe Kwon**
MIT

**Mantas Mazeika**
UIUC

**Richard Li**
UC Berkeley

**Dawn Song**
UC Berkeley

**Jacob Steinhardt**
UC Berkeley

**Owain Evans**
University of Oxford

**Dan Hendrycks**
UC Berkeley

## A   Additional Experimental Details and Results

### A.1   Autocast Experiments

**Calibration Results.**   In Figure 2, we show the adaptive binning calibration curve for crowd forecasts on all resolved true/false questions by plotting the fraction of positives against the model's predicted probability for the positive class.

Additionally, we can compare the calibration of our static and temporal models to crowd performance on the resolved test set. Treating true/false questions as two-class classification problems and combining them with multiple-choice questions, we calculate adaptive binning calibration error with a bin size of 50 samples. The largest FiD Static model incurs a $40\%$ calibration error while the human crowd incurs a much smaller $8\%$ calibration error. By leveraging crowd predictions in our FiD Temporal models, we reduce the calibration error to $17\%$, showing potential for improvements.

**Model and Training Loss.**   The FiD Temporal model uses three separate linear heads after its hidden state outputs to answer each type of questions (true/false, multiple-choice, and numerical). In particular, the multiple-choice head has 12 outputs which is the maximum number of choices in the training set. Additionally, the original input embeddings are replaced with a linear layer to map from the FiD Static's hidden states to the GPT-2's hidden states. Finally, to make training more stable, we average the loss over the sequence of predictions for each question to weigh the questions evenly. Moreover, the losses of the three types of questions are normalized by their respective baseline loss (uniformly random predictions) before summing together so that their losses are on the same scale.

**Retrieval from CC-NEWS.**   Given a question, for each day the question is active, we retrieve the top 10 relevant news articles from the daily articles. In our FiD-Temporal experiments, we only use the top 1 from every day. Then, we aggregate all these articles from different dates and rank them according to the retrieval score. The top 10 articles are used for the FiD-Static model. We follow the Terms of Use for the Common Crawl website. The dataset is fully reproducible with the script to download and filter CC-NEWS on GitHub.

### A.2   Confidence Intervals

**Interval Construction.**   In the reference implementation of the get_confidence_intervals function in Figure 1, we construct our intervals by first producing a point estimate for each question and

```python
Is = [0.5, 0.55, ..., 0.95]
num_intervals = len(Is)

def low_containment_mask(lowers, uppers, labels, Is):
    # lowers, uppers: Predicted lower and upper bounds of intervals
    # Is: Target confidence levels
    # Returns: A list of boolean values indicating which confidence level
    #          has containment ratio below the target level within batch
    contained = (lowers <= labels) * (labels <= uppers)
    ratio_contained = contained.mean(dim=0)
    return ratio_contained < Is

def get_confidence_intervals(logits):
    # logits: Model output with (2 * num_intervals + 1) neurons
    deltas, point_estimates = softplus(logits[:, :-1]), logits[:, -1:]
    lower_deltas = deltas[:, :num_intervals]
    higher_deltas = deltas[:, num_intervals:]
    interval_lengths = lower_deltas + higher_deltas
    # custom cumsum with gradients accumulated once on each delta
    lower_deltas = utils.cumsum(lower_deltas)
    higher_deltas = utils.cumsum(higher_deltas)
    lowers = point_estimates - lower_deltas
    uppers = point_estimates + higher_deltas

    return lowers, uppers, point_estimates, interval_lengths

out = get_confidence_intervals(logits)
lowers, uppers, point_estimates, interval_lengths = out

𝓛_p = MSE(point_estimates, labels)
l_mask = lowers > labels
u_mask = uppers < labels
𝓛_b = MSE(lowers, labels) * l_mask + MSE(uppers, labels) * u_mask
𝓛_i = MSE(interval_lengths, 0)
# normalize loss by the label magnitude, adjusting for small labels
𝓛_p /= (1 + abs(labels))
𝓛_b /= (1 + abs(labels))
𝓛_i /= (1 + abs(labels))
# activate loss for particular confidence levels based on ci_mask
ci_mask = low_containment_mask(lowers, uppers, labels, Is)
𝓛_b = 𝓛_b.mean(dim=0) * ci_mask
𝓛_i = 𝓛_i.mean(dim=0) * (1 - ci_mask)

α, β, γ = 1, 1, 0.01 # hyperparameters
loss = α * 𝓛_p.mean() + β * 𝓛_b.mean() + γ * 𝓛_i.mean()
```

Figure 1: A reference implementation of the baseline training loss for outputting calibrated confidence intervals. For the confidence levels where too few true labels fall inside the predicted intervals, we encourage the model to adjust its boundaries through boundary loss $\mathcal{L}_b$. Conversely, we encourage the model to shrink the predicted intervals if too many fall inside the predicted intervals.

iteratively adding on non-negative, non-symmetric deltas on both sides, so that the intervals become nested and wider for higher confidence levels.

**Training Loss for Baseline.**  First, because the labels span a large numerical range, we normalize them by taking the $log$. Then, we construct a loss with three components shown in Figure 1: (1) $\mathcal{L}_p$: MSE loss between the predicted point estimate and the ground-truth target, (2) $\mathcal{L}_b$: MSE loss between the boundaries of the predicted confidence intervals and the ground-truth target for boundaries that are on the wrong side of the target, (3) $\mathcal{L}_i$: a penalty on the length of the predicted intervals to encourage

**Algorithm 1** RMS Calibration Error
___

1: **Input:** A set of $N$ examples each with label $\{y_i\}_{i=1}^N$ and $C$ predicted confidence intervals $\{(l_i^c, u_i^c)\}_{c=1,i=1}^{C,N}$ corresponding to $C$ confidence levels $\{\mathcal{I}^c\}_{c=1}^C$ (e.g., $\mathcal{I}^C = 0.95$). Set bin size to $M$.

2: **function** AdaptiveRMS

3:     Sort the examples by labels $y_n$ in ascending order.

4:     Assign a bin label $b_k = \lfloor \frac{k-1}{M} \rfloor + 1$ to each by splitting sorted examples into chunks of $M$.

5:     Let $\{B_i\}_{i=1}^b$ be the set of bins and $B_i$ the subset of examples in bin $i$.

6:     **for** $c = 1, \ldots, C$ **do**

7:         Calculate empirical containment for bin $i$

$$\widehat{p}_i^c = \frac{1}{|B_i|} \sum_{k \in B_i} \mathbb{1}(y_k \in [l_k^c, u_k^c])$$

8:         Calculate root mean squared calibration error

$$\mathrm{RMS}^c = \sqrt{\frac{1}{b} \sum_{i=1}^b (\widehat{p}_i^c - \mathcal{I}^c)^2}$$

9:     **end for**

10:     Output $\frac{1}{C} \sum_{c=1}^C \mathrm{RMS}^c$

11: **end function**
___

finer predictions. Based on whether the ratio of true labels contained in the predicted intervals is higher than the target confidence level, we either activate the boundary loss $\mathcal{L}_b$ or the interval length loss $\mathcal{L}_i$ for that particular confidence level output. Lastly, the three components are weighted by coefficients $1, 1, 0.01$ chosen with a simple search using the validation set.

**Adaptive RMS Metric.** An important task for numerical forecasting is outputting calibrated uncertainty estimates. However, a unique challenge in this setting is that answers can vary across many orders of magnitude. To evaluate the calibration of confidence intervals across a large dynamic range of output values, we design a specialized local calibration metric (Zhao et al., 2020; Kull et al., 2019), shown in Algorithm 1. First, test examples are sorted by their target value and split into bins with a fixed number of examples each (adaptive binning). Then, we calculate calibration error across all bins using a Euclidean norm (Hendrycks et al., 2019). Finally, we average this local calibration error across all confidence levels, giving the final metric. For brevity, we refer to this overall metric as RMS Calibration Error. A low value for this error metric indicates that models are calibrated across their entire dynamic range of output values.

|  | Resolved | Unresolved | Total |
|---|---|---|---|
| Train | 2815 | 1375 | 4190 |
|  | 4411 | 1974 | 6385 |
| Test | 907 | 1610 | 2517 |
|  | 1292 | 2305 | 3597 |
| Total | 3722 | 2985 | 6707 |
|  | 5703 | 4279 | 9982 |

Table 1: The number of forecasting questions in Autocast. In total, there are nearly 10,000 questions. Gray text indicates the number of questions after augmenting true/false questions with their negations, a procedure we use to balance the dataset.

**Calibration Dataset Statistics.** The dataset of numerical questions gathered for our calibration evaluations has training, validation, and test sets containing 32,200, 3,443, and 6,170 examples respectively.

# B    Additional Dataset Information

## B.1    Dataset Details

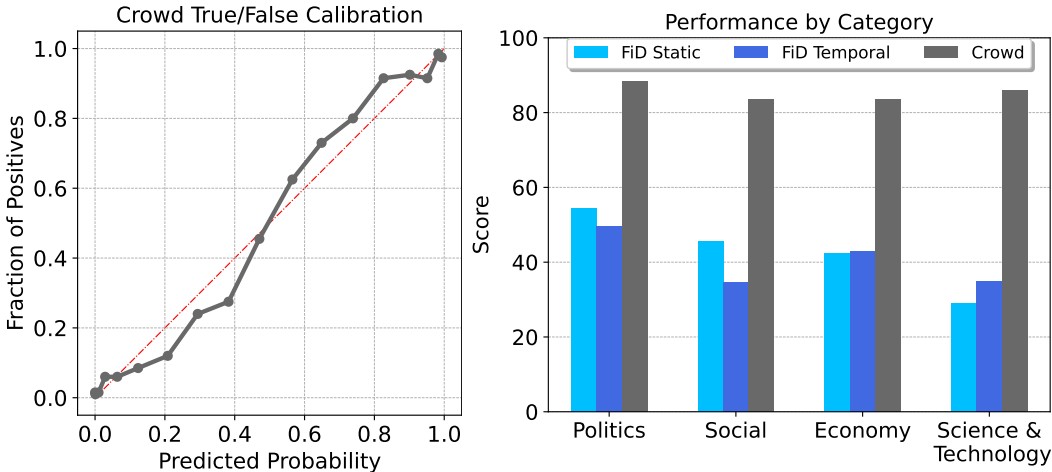

Figure 2: Left: Crowd forecasts for true/false questions have good calibration. Right: The per-category performance of baselines. Score indicates the combined score metric.

The Autocast dataset contains 6,707 unique questions in total, spanning three question types, including resolved and unresolved. After we balance the true/false questions by adding negated questions, the true/false question count doubles, making the grand total 9,757. The numbers of training and test examples are shown in Table 1 for ease of reference. The numbers below are based on the expanded dataset using true/false balancing. The Autocast training set

|       | T/F  | MCQ | Numerical |
|-------|------|-----|-----------|
| Train | 3187 | 753 | 471       |
| Test  | 775  | 176 | 341       |
| Total | 3962 | 929 | 812       |

Table 2: The number of resolved questions in Autocast, grouped by question type.

we experiment with does not include unresolved questions. This training set contains 4,411 examples, and the test set contains 1,292 examples. To prevent leakage of future information, the train set consists of all questions that closed or resolved before 5-11-2021 and the test set consists of all questions that closed or resolved after 5-11-2021. In addition, we also release 1,974 unresolved train questions having a publish date before 5-11-2021 and 2,305 unresolved test questions published after 5-11-2021. Note that our baselines do not use any unresolved questions, so there is a guarantee of no leakage. However, training with auxiliary training signals from unresolved questions (e.g., crowd forecasts) requires additional care to ensure no leakage. Namely, crowd forecasts from after 5-11-2021 must not be used.

**Per-Category Performance.** In Figure 2, we show performance by category using the combined score metric. Science & Technology questions are the most challenging for the FiD Static and FiD Temporal baselines, while the crowd predictions perform similarly on all question categories. There is a substantial gap between models and crowd predictions, but crowd predictions are still far from a perfect score of 100%.

**Computation of Crowd Forecasts.** The human crowd forecasts are directly obtained from forecasting platforms, and the precise meaning depends on the platform. For example, for Metaculus questions the crowd forecast represents the median forecast with the recent player predictions weighted more. For Good Judgment Open questions, it represents the median of the recent 40% of forecasts. In all cases, the crowd forecast aggregates previous individual forecasts at a given time.

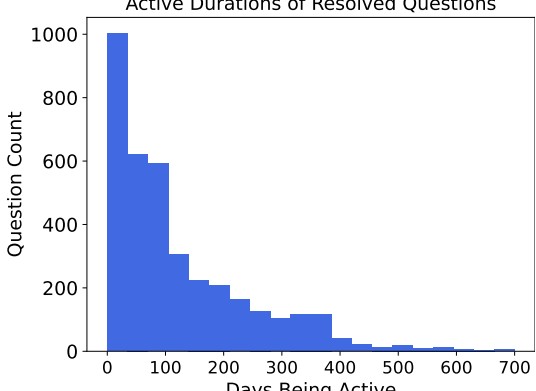

Figure 3: We visualize the distribution of the duration of the active periods for Autocast questions. Questions vary greatly in terms of how long they are active in the forecasting market, with questions taking up to years to resolve.

| Category | Percentage | Subcategories |
|---|---|---|
| Politics | 31% | Geopolitics, Security and Conflict, Elections, Foreign Policy, Leader Entry/Exit, Law, Economic Policy, US Policy, Ukraine |
| Social | 22% | COVID-19, Social Issues, Environment, Effective Altruism, Sports, Entertainment, Health, Society, Pandemic, Animal Welfare, Metaculus, Climate, Education |
| Science & Tech | 21% | Technology, Computing, Biological Sciences, Physical Sciences, Computer Science, Biology, Human Sciences, AI, Mathematics, Tech |
| Economy | 20% | Business, Finance, Industry, Economic Indicators, Infrastructure, Microelectronics, Semiconductors |
| Other | 6% | Other, Open |

Table 3: The percentage of Autocast questions in each category, and the subcategories belonging to each category. Autocast questions have fairly even coverage of a wide variety of topics.

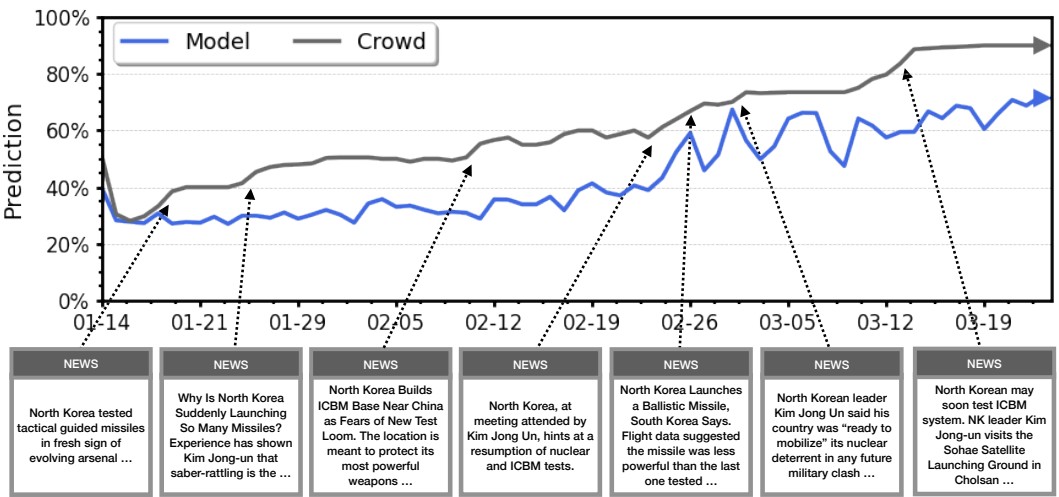

Figure 4: The same example from the Autocast dataset shown in the main paper, illustrating how the crowd forecast is influenced by news articles published throughout the prediction period.

## B.2 Legal Compliance.

We scrape content from public forecasting sites to build Autocast. In some cases, this data may be protected under copyright. We received full legal permission from Metaculus for our use case, and in all cases we abide by Fair Use §107: "the fair use of a copyrighted work, including such use by ... scholarship, or research, is not an infringement of copyright", where fair use is determined by "the purpose and character of the use, including whether such use is of a commercial nature or is for nonprofit educational purposes", "the amount and substantiality of the portion used in relation to the copyrighted work as a whole", and "the effect of the use upon the potential market for or value of the copyrighted work." Autocast is meant for academic, non-commercial use only and only scrapes publicly visible data from the websites, which excludes individual forecasts.

### B.3 Author Statement and License.

We bear all responsibility in case of violation of rights. Parts of the Autocast data may be under copyright, so we do not provide an official license and rely on Fair Use §107. Our code is open sourced under the MIT license.

# C    X-Risk Sheet

We provide an analysis of our paper's contribution to reducing existential risk from future AI systems following the framework suggested by (Hendrycks and Mazeika, 2022). Individual question responses do not decisively imply relevance or irrelevance to existential risk reduction.

## C.1    Long-Term Impact on Advanced AI Systems

In this section, please analyze how this work shapes the process that will lead to advanced AI systems and how it steers the process in a safer direction.

1. **Overview.** How is this work intended to reduce existential risks from advanced AI systems?

   **Answer:** This work builds towards improving institutional decision making and systemic safety. In short, this could help resolve matters of fact that influence policies and decisions made by political leaders in an increasingly complex modern world, putting humanity in a better place to deal with the global turbulence and uncertainty created by AI systems when they rapidly reshape society. A fuller motivation for "ML for Improving Epistemics" is described in Hendrycks and Mazeika (2022).

2. **Direct Effects.** If this work directly reduces existential risks, what are the main hazards, vulnerabilities, or failure modes that it directly affects?

   **Answer:** This directly works against failure modes such as eroded epistemics and hazards such as highly persuasive or manipulative AI systems.

3. **Diffuse Effects.** If this work reduces existential risks indirectly or diffusely, what are the main contributing factors that it affects?

   **Answer:** This work could lead to improved decision making, epistemics, and collective intelligence. Automated forecasting tools could eventually assist various levels of the sociotechnical hierarchy, including congress and legislatures; government regulatory agencies, industry associations, user associations, etc.; and company management. This lowers the risk of conflict that would accelerate the weaponization of AI, so it diffusely works against weaponized AI failure modes.

4. **What's at Stake?** What is a future scenario in which this research direction could prevent the sudden, large-scale loss of life? If not applicable, what is a future scenario in which this research direction be highly beneficial?

   **Answer:** Advanced automated forecasting better enables political leaders to avoid precarious moments that could spark a large-scale conflict.

5. **Result Fragility.** Do the findings rest on strong theoretical assumptions; are they not demonstrated using leading-edge tasks or models; or are the findings highly sensitive to hyperparameters?    ☐

6. **Problem Difficulty.** Is it implausible that any practical system could ever markedly outperform humans at this task?    ☐

7. **Human Unreliability.** Does this approach strongly depend on handcrafted features, expert supervision, or human reliability?    ☐

8. **Competitive Pressures.** Does work towards this approach strongly trade off against raw intelligence, other general capabilities, or economic utility?    ☐

## C.2    Safety-Capabilities Balance

In this section, please analyze how this work relates to general capabilities and how it affects the balance between safety and hazards from general capabilities.

9. **Overview.** How does this improve safety more than it improves general capabilities?

   **Answer:** While this line of work reduces systemic risk factors and can improve institutional decision making, making AI systems better at forecasting could potentially improve general capabilities. Its relation to general capabilities is currently unclear. In humans, at the extremes, IQ is hardly predictive of forecasting ability, suggesting forecasting of near-term geopolitical events is a specific and not general skill. Likewise, work in this space could focus on engineering better forecasting systems rather than improving general representations, so as to avoid capabilities

externalities; this is potentially a more robust strategy for avoiding capabilities externalities. If it turns out that capabilities externalities are difficult to avoid even while simply engineering better forecasting systems, we would suggest that safety researchers stop working on this problem.

10. **Red Teaming.** What is a way in which this hastens general capabilities or the onset of x-risks?

    **Answer:** Making AI systems better at forecasting could also improve general capabilities or at least the raw power of AI systems. As Yann LeCun reminds us, "prediction is the essence of intelligence."

11. **General Tasks.** Does this work advance progress on tasks that have been previously considered the subject of usual capabilities research? ☐

12. **General Goals.** Does this improve or facilitate research towards general prediction, classification, state estimation, efficiency, scalability, generation, data compression, executing clear instructions, helpfulness, informativeness, reasoning, planning, researching, optimization, (self-)supervised learning, sequential decision making, recursive self-improvement, open-ended goals, models accessing the internet, or similar capabilities? ☒

13. **Correlation With General Aptitude.** Is the analyzed capability known to be highly predicted by general cognitive ability or educational attainment? ☐

14. **Safety via Capabilities.** Does this advance safety along with, or as a consequence of, advancing other capabilities or the study of AI? ☒

## C.3 Elaborations and Other Considerations

15. **Other.** What clarifications or uncertainties about this work and x-risk are worth mentioning?

    **Answer:** Regarding Q7, while human forecasters are important for building a training set with rich annotations, the actual human forecasts are unnecessary, as technically only the resolutions are needed. Additionally, the end goal is to create automated forecasting systems that do not depend on human reliability. Eventually, these systems could become much faster and more reliable than human forecasters.

    Regarding Q12, this work facilitates research towards general prediction of future events and consequently toward improved planning. However, we expect the kinds of predictions improved by forecasting research to be especially relevant for reducing x-risk. For example, improved institutional decision making surrounding geopolitical events could reduce the risk of global conflicts leading to the weaponization of strong AI.

    Regarding Q13, IQ is predictive of forecasting ability in humans, not overwhelmingly so (Mellers et al., 2015). Moreover, its correlation is especially weak at extremes. Likewise, forecasting skills for near-term geopolitical events are partly learnable, further suggesting a separation from general cognitive ability.

    Regarding Q14, while the relationship between general capabilities and research on forecasting near-term geopolitical events is currently unclear, this research does advance the study of narrow AI systems.

    Finally, we would like to discuss limitations and potential hazards of relying on ML for forecasting near-term geopolitical events.

    (a) Forecasting is best used for refining understanding rather than for anticipating the future more generally. Forecasters are demonstrated to be useful for optimizing probabilities for somewhat likely events (e.g., events with probabilities between, say, 5% and 95%). What is more important are tools that unearth important considerations that were implicitly assigned negligible probabilities or wrongly treated by humans as misinformation or worth ignoring. These considerations are often not forecasted and are not thought worth asking; implicitly, such events could the thought to be assigned low probabilities (e.g., say $10^{-7}$), while some people argue that these considerations are more likely than others believe (e.g., say $10^{-1}$). The information value provided from putting ignored considerations on our radar is substantial, in fact, orders of magnitude greater than the information gained by refining probabilities by a few percent. Forecasting competitions are about refining estimates of known unknowns–questions already on our radar–but what is better for risk reduction is confronting unknown unknowns, finding considerations to put on our radar, and reducing *exposure* to inchoate potential risks. For this reason, Hendrycks et al. (2021) suggest tools that improve brainstorming and suggesting considerations.

(b) Forecasting is not necessarily a suitable tool for addressing tail risks. Taleb and Tetlock (2013) remind us that "No one has yet figured out how to design a forecasting tournament to assess the accuracy of probability judgments that range between .00000001% and 1%—and if someone ever did, it is unlikely that anyone would have the patience–or lifespan–to run the forecasting tournament for the necessary stretches of time (requiring us to think not just in terms of decades, centuries and millennia)." Taleb and Tetlock (2013) further remind us that it is unjustified to use forecasting tools for revolutions, market crashes, venture capital, or other winner-take-all domains. Furthermore they note that framing questions about tail risks as "a binary question is dangerous because it masks exponentially escalating tail risks." Consequently, "improving short-run probability judgments" and "contingency planning for systemic [tail] risks" are "complementary" and separate (Tetlock et al., 2022). Indeed, superforecasters usually anchor in outside view (Tetlock and Gardner, 2016), which neglects systemic risks. In environments with tail events, it is not how often one is correct that matters but rather how large one's cumulative errors are; current forecasting metrics do not sufficiently penalize forecasters that ignore tail risks nor do they greatly reward prescience about Black Swans.

(c) Forecasting tools could lead to risky behavior. For example, forecasting systems may induce inaction. If forecasts are uncertain, leaders may argue that "we should not make a decision before we have a reliable forecast" so we should "sit tight and assess." This is sometimes referred to as the delay fallacy, namely "if we wait we will know more about X, hence no decision about X should be made now" (Hansson, 2004). However, it is often cheaper to prevent risks or reduce exposure to risks, as "an existential risk needs to be killed in the egg, when it is still cheap to do so" (Taleb et al., 2020). Waiting until all the relevant information arrives is often waiting until it is too late.

Furthermore, humans are known to misinterpret probabilities (Vodrahalli et al., 2022). Systems that assign an event 3% probability may lead decision-makers to assume the event will not happen. Automation bias may mean forecasting systems induce users to have a gain in confidence that is greater than their gain in knowledge. Risk compensation suggests this could result in riskier actions (Hedlund, 2000). Furthermore, forecasts are often not provided with reverse psychology in mind. However, a forecasting system that forecasts a low risk can lead users to act as though there is no risk and increase risky behavior, which increases systemic risk.