# OpenReview forum: "Forecasting Future World Events With Neural Networks"
_NeurIPS.cc/2022/Track/Datasets_and_Benchmarks — NeurIPS 2022 Datasets and Benchmarks _

### Official Review · Reviewer_RFE7 · 2022-07-02
**A nice forecasting question dataset accompanied by a news article corpus (not released yet) with date information and fair benchmarks**

**Rating:** 7
**Confidence:** 3
**Clarity:** The paper is well-writen.

**Strengths:**

### Significance of the contribution

The Autocast dataset along with the news corpus have the following features, which, as a combination, is unique.
1. Forecasting questions are professional, i.e., coming from three well-known public forecast tournaments (GJP, Metaculus, and CSET Foretell)
2. Forecasting questions and news articles have date information so that they allow ML models to simulate the real forecasting environment and avoid information leakage.

### Relevance to the broader research community
I believe forecasting future events is an important topic that have many real-world applications. The provided dataset will help the general forecasting community as well as the machine forecasting community.

### Accessibility and accountability
Autocast is easily accessible on Github, with most fields clearly defined.



**Weaknesses:**

### Significance of the contribution

It seems that the dataset and the news corpus are all tailed from existing datasets and projects. However, the effort should still be meritted.

### Accessibility and accountability
* The accompanying news corpus dataset is not released according to the github page.

* Resolved date is missing in the dataset, which I believe is very important as forecasts only make sense before resolution and the resolved date may be earlier than the given close date. In other word, the information leakage problem is not entirely solved if the resolved date is not provided.

* The human crowd forecasts is an important feature as well as a good target to learn in this dataset. However, the authors did not clarify how the human crowd forecasts are generated or obtained.

* It would be better if the authors introduced the distribution of the time scope of the forecasting questions, e.g., how many of them are short-term questions, e.g., taking weeks to resolve, and long-term questions, e.g., taking one or two years to resolve.

**Additional Feedback:**

Seen questions left in the above aspects.

**Correctness:**

The forecasting question dataset Autocast is constructed in a sound way. The forecasting questions are from three serious forecast tournaments and most of the important information, including the open date, close date and resolution status. The questions are well-defined and covered a wide range of important topics and geographic areas. The authors also added negated questions to balance the ground truth labels. A small question is that why the authors added these negated questions instead of replacing the original questions, as adding negated questions may change the distribution of the original questions.

I believe that the benchmark evaluation is well-conducted. The authors used percent accuracy as the metric, which is intuitive and straightforward. However, for evaluating forecasting for T/F for multiple-choice questions, the proper scoring rules, especially the Brier score (MSE) are the most widely used metric, which also evaluate the calibration of forecasts. I was wondering why the authors did not try this metric.

**Documentation:**

Autocast is a well-documented, except for the following aspects:
1. How the human crowd forecasts are obtained
2. The original source of the each forecasting question, i.e., whether the question is from GJP, metaculus, or CSET Foretell

The news corpus seems not released yet.
The benchmarks and the evaluation are well documented.

**Ethics:**

No concern.

**Relation To Prior Work:**

The related work is properly discussed.

**Summary And Contributions:**

The authors introduced a judgmental forecasting question dataset Autocast (accessible) along with a news corpus (not released yet) with date information for the purpose of facilitating the research of using language models to forecast future events. The authors also provided several benchmarks constructed using various popular language models and time-series models as well as human crowd predictions. The benchmarks performance also revealed that retrieving relevant articles could help improve the performance of machine forecasts, while there still existed a large improvement space for machine forecasts when compared to human crowd forecasts. Although there are several existing mature forecasting datasets, Autocast along with the news corpus with date information is still a progress as they provide an environment for language models to simulate the real forecast environment (if the news corpus is released and can be verified).

---

> ### Author Response · Authors · 2022-08-16
> **Response to Reviewer RFE7**
>
> Thank you for your careful analysis of our work. We hope the following response addresses your concerns.
>
> **News Corpus Is Now Available.** The Common Crawl news corpus is now linked to in the GitHub page.
>
> **Resolve Date.** The resolve date is referred to as close_time value in the annotations. This is because all questions are published with a close date (a date after which the question is closed and new predictions are not allowed), and the resolve date is distinct from the close date. For example, a question concerning an event in 2030 may stop accepting new predictions in 2025, but it will not resolve until 2030. The close_time value is the minimum of the close date and the resolve date and represents the point at which human forecasters were no longer allowed to make predictions. We use this date to determine which articles are available for retrieval, which puts the models on equal footing with human forecasters for each question. We will clarify this in the GitHub repository.
>
> **Crowd Forecasts.** The human crowd forecasts are directly obtained from forecasting platforms, and the precise meaning depends on the platform. For example, for Metaculus questions the crowd forecast represents the median forecast with the recent player predictions weighted more. For Good Judgment Open questions, it represents the median of the recent 40% of forecasts. In all cases, the crowd forecast aggregates previous individual forecasts at a given time. We have updated the paper to include this information thanks to your suggestion.
>
> **Distribution of Question Time Scope.** We have updated the paper to include the distribution of close dates (dates at which the question will stop accepting new predictions) and resolution times (time between the opening and resolution dates for resolved questions). This is in Figure 9 of the Appendix. Many resolved questions take a long time to resolve, and there are some questions that will continue accepting submissions decades into the future. Thus, Autocast contains questions about near-term and long-term events.
>
> **Original source of question.** Each question has a unique id. The first character denotes the forecasting site. The number after the character denotes the question id on the forecasting site. For example, M1 is question 1 from Metaculus, which corresponds to https://www.metaculus.com/questions/1. We have added this to the documentation. Thank you for your suggestion.
>
> **Reason for Adding Negated Questions.** We add negated questions to ensure that random performance on the true/false questions is 50% (i.e., to balance the dataset). For this to work, we need to keep the original questions as well. We could have replaced a portion of the original questions with negated questions to the same effect, but we also wanted to retain all of the original questions in the final dataset. Hence our solution of adding the negated questions to the dataset.
>
> **Brier Score.** We decided to use accuracy in our evaluations, because the Brier score combines prediction error and calibration error, and we wanted to isolate these effects. As we show in the Appendix, current models have a very high calibration error on Autocast, so the Brier score is dominated by this high calibration error. Post-hoc calibration techniques could be used to improve calibration and render the Brier score a less noisy metric, and we are investigating this for future work.

---

> > ### Comment · Reviewer_RFE7 · 2022-08-29
> > **Response acknowledgement**
> >
> > Thanks for the authors' responses. My concerns are well-addressed. I would recommend the authors to contain a brief discussion about the performance in Brier score, e.g., in appendix for reference. Except the ethic concern, which is not my expertise, I would like to see this article published in this venue and would keep my original score.

---

### Official Review · Reviewer_tdsv · 2022-07-21
**A dataset composed of human forecasting tournaments questions and a curated news text corpus (date sorted, to be released soon), as benchmark for learning future world events forecasting models**

**Rating:** 7
**Confidence:** 4

**Strengths:**

The presented work deals with important and challenging question of constructing datasets that can be used to test the ability of learning algorithms to build models that forecast future world events from large corpora of text data. Authors take care that training procedure and train/test split respect the notion of forecasting problem and cleanly separate past and future data. Provided baselines check dataset sanity and show clear gap to human crowd performance, providing motivation to use the dataset for further development and improvement in this important direction.

**Weaknesses:**

One concern in my opinion is a small scale of the questions dataset. Few thousands of those may by far not be conclusive enough to evaluate performance of state-of-the art algorithms at scale.

Further limitation (minor, as it can be readily lifted) is in my eyes decision to use only news text corpora for training. Arguably, other available sources of information (eg stock exchange behavior, weather, text corpora like Wikipedia dated before the relevant events to forecast) may be helpful to enhance learning.

**Additional Feedback:**

- Would it be possible to extend questions dataset in semi-automated manner by taking some of the already available questions and rephrasing those in different time context (for instance, Will X become a president in Y year can be re-used for multiple time points and multiple person candidates; same procedure applied to other topics may help to expand the dataset profoundly)

**Clarity:**

The paper provides good motivation for the subject of building datasets for benchmarking algorithms that learn future world events forecasting models and is well and clearly written.

**Correctness:**

The claims put forward by authors have sound back up in the conducted experiments.

**Documentation:**

The authors provide sufficient info for the constructed dataset and also deliver dataset sheet in the supplementary.

**Ethics:**

The authors elaborate in detail on ethical concerns and provide a dedicated X-Risk sheet in the supplementary material that illuminates paper's contribution to reduction of existential risk from future AI systems.

**Relation To Prior Work:**

Prior work is properly addressed and discussed thoroughly.

**Summary And Contributions:**

The submission provides clear motivated and well written contribution to establishing datasets that test the ability of learning algorithms to build models forecasting future world events of importance from available text news corpora. The authors gather the dataset (Autocast) comprising 6707 questions by making use of three public forecasting tournaments (Metaculus, Good Judgment Open and CSET Foretell) that feature human forecast experts crowd. Questions contain true/false, multiple choice and numerical ground truth answers. Text news corpora is curated from Common Crawl and is organized by date, allowing to mimic the conditions under which humans make past forecasts, avoiding leakage from the future information.

Authors further present baseline approaches for learning forecast models from the presented dataset. They compare approaches without retrieval (based on zero-shot UnifiedQA-v2 and T5 fine-tuning) and with retrieval (based on BM25 for retrieval and on Fusion-in-Decoder, FiD, for encoding retrieved news articles). The retrieval-based method is either considering final output and top-10 ranked retrieved articles (FiD static) or takes into account human crowd predictions evolving in time between start and closing of a question (FiD temporal, using GPT-2 autoregressive model for intermediate forecast outputs).


Comparing baseline methods to human crowd performance, authors observe advantage for retrieval-based methods (FiD static, FiD temporal) against non-retrieval ones (that perform close to random guess level on the task). The observed gap to the human performance is substantial, which make authors call for further development of forecast learning algorithms. The authors do not observe substantial difference between FiD static and FiD temporal retrieval-based methods, with the conclusion that further efforts have to be undertaken to make better use of auxilliary losses for the forecasting task.

Further, the authors  provide an additional dataset to calibrate output of the forecasting models on numerical answer, where they source the questions from NLP datasets covering diverse topics and with answers varying across orders of magnitude. They observe that with increasing model size (using DeBERTa-v3 model backbone), calibration improves along with other performance metrics.

---

> ### Author Response · Authors · 2022-08-16
> **Response to Reviewer tdsv**
>
> Thank you for your careful analysis of our work. We hope the following response addresses your concerns.
>
> **Number of Questions is Similar to Other Datasets.** To increase the number of questions and improve class balance, we negate all the true/false questions in Autocast using the GPT-3 edit feature (with manual quality assurance). After true/false balancing, the total number of questions in Autocast is 9982. This is similar to the number of questions in other datasets developed for fine-tuned models (e.g., MATH, MMLU, LogiQA, ForecastQA). Additionally, the number of questions in Autocast will grow considerably larger over time in future versions of the dataset (see Fig. 2 for evidence that the number of questions submitted to forecasting tournaments is rapidly increasing).
>
> **Using Other Sources of Data.** We agree that using other sources of data besides news articles could improve performance, and we think this is an exciting direction for future work.
>
> **Extending Questions.** It should be possible to increase the number of questions using the idea you describe. Currently, this would require manual work, since the questions do not follow precise templates, but it is certainly an interesting idea. Oftentimes questions are submitted to the forecasting tournaments that are variations on each other, e.g., “will a study have outcome X”, “will a study have outcome Y”, etc.

---

> > ### Comment · Reviewer_tdsv · 2022-08-28
> > **Response acknowledgement**
> >
> > I would like to thanks the authors for their response. My questions were fully addressed. I would recommend the authors to include the discussion on dataset scale and its future development they provided in short form in the response, as this may give hints to the community how the exisiting dataset compares to related efforts and what will be the tendency for scaling it up.
> > The ethics review raised concerns on copyright matters which I am not expert in, therefore I am keeping the original score.

---

### Official Review · Reviewer_RYpB · 2022-07-23
**An interesting and challenging dataset for forecasting**

**Rating:** 7
**Confidence:** 4

**Strengths:**

1.  The problem that the authors address is of great importance. Forecasting future world events can be helpful in a variety of domains.
2.  The authors show that the state-of-the-art language models are significantly worse than the human baseline, and this dataset can open up a great opportunity for researchers to improve the language models for forecasting.
3.  The paper is very well written and details are explained.
4.  This dataset is more accurately collected and is more realistic compared to prior works.
5.  The authors take the necessary precautions to avoid leakage from the training set to the test set.


**Weaknesses:**

Most of the weaknesses are minor.

1.  In the "Metrics" section, it is worth mentioning that for T/F and MCQ metrics the higher is better, while for the numerical metric the lower is better to avoid confusion for readers. Also, in the combined score metric which is (T/F + MCQ - Numerical)/2, why is the denominator two instead of three?
2.  Are collected news from CommonCrawl only used for training or do you use them for the test as well? Also, are they only used for retrieval-based models? Please clarify these and state them in the paper.
3.  Would you please clarify why you use the top 1 article in the FiD Temporal and top 10 articles in the FiD Static? That does not seem to be a fair comparison.
4.  On lines 209-211 you say that you limit the article length to 512 tokens? Please specify whether you consider "tokens" to be words or characters.
5.  In table 2, column Numerical, why 19.5 is bold given that it is not the max?
6.  X axis in figure 1 does not have a label. Although it can be inferred by the reader, it would make it more clear to indicate that the X axis is the date. Also, is the prediction percentage for model (blue line) an average of multiple models? Since the graph is the performance for a single binary question, I assume it is an average of multiple models, but it needs to be clarified what models.
7. Additional graphs similar to figure 4 illustrating the FiD static model and the two models without retrieval would be helpful for understanding the differences among all these models.


**Additional Feedback:**

1.  Are there any open-ended questions in the dataset such as "who is more likely to win the next Nobel prize?" where the answer is not numerical and is not also limited to multiple choices to choose from? If yes, please mention it in the paper. If not, why did you choose not to have them, and do you have any plans to add them?
2.  On line 138, you mention that your dataset also includes metadata. Would you please state whether you use these metadata for training or not?
3.  Do you have any insights or hypothesis as to why the performance decreases for numerical questions when you increase the number of parameters of both FiD models?
4.  I understand your argument regarding leakage from training to test if we have access to information/news after the fixed date in the training set. However, I am confused by the following statement on lines 40-42. "When it comes to learning, humans cannot be trained on past data (e.g. How likely was the Soviet Union’s collapse from the viewpoint of 1980?) because they know the outcomes – but past data can be used for ML models." Humans study history and can be trained (in a supervised manner) using past data. Also, data by definition is something that has already happened (in the past), and the term "past data" is confusing to me. With that said, I guess maybe you meant that humans cannot be **evaluated** on predicting events that have already happened, which I agree with. Please correct me if I am wrong.

**Clarity:**

The paper is very easy to read and keeps the reader engaged. The methodology is clearly explained for most parts. Refer to "Weaknesses" section for more details.

**Correctness:**

The dataset is constructed in a correct method. The evaluations are thorough and valid. The only concern is mentioned in item three of the "Weaknesses" section.

**Documentation:**

The authors provide the dataset and the code partially.
The accompanying news articles are not provided yet. Also, I could not find the code for models without retrieval in the GitHub repository.
The details/plans of updating the dataset is not provided.

**Ethics:**

This paper uses publicly available data and models, therefore it does not introduce any ethical concerns on its own unless it stems from previous works.
The authors state that they abide by Fair Use §107 in the supplementary material on line 78. The authors also provide a datasheet for their dataset answering a set of questions related to ethics.


**Relation To Prior Work:**

The authors position their work very well with respect to different works related to the topic of forecasting including calibration, information retrieval, and forecasting itself.

**Summary And Contributions:**

The authors propose a dataset of questions and answers for predicting world events. The dataset includes the questions, the start and close dates, the answers (if the questions have been resolved), the time-series of crowd forecasts by human experts, and an accompanying news corpus sorted by date collected from CommonCrawl. The authors state that they intend to update the dataset regularly with new questions and resolutions.

This dataset has diverse question types including multiple choice questions (MCQ), True/False questions (T/F), and numerical questions. The dataset covers diverse topics such as politics, economics, society, science, and a category for everything else called *other*, and altogether they have a total of 44 subcategories. The questions have different time horizons; some questions are resolved while some of them are still unresolved. The dataset is collected from public human forecasting tournaments and amounts to 6,706 questions in total. The authors balance the True/False questions by adding negated questions using GPT-3 and then manually checking for correctness. Moreover, they include an auxiliary dataset of numerical questions that includes about 30,000 questions. This auxiliary dataset provides the groundwork for measuring calibration.

Train/test split is done based on the resolution date of questions so that questions in the test set all resolve after a specific date, and the training set includes the questions that resolve before that specific date.
The authors prevent the models from overfitting/memorization by using only those news articles that were available before the resolution date of a forecast.

The authors train two sets of models. The first set is models without retrieval including UnifiedQA-v2 and T5, and the second set is retrieval-based models including *FiD static* and *FiD temporal* that use news articles. FiD static uses the top 10 retrieved articles, while the FiD Temporal uses the top 1 article and in addition leverages the intermediate crowd predictions (before the question is resolved) as auxiliary supervision. Their results show that their temporal method neither harms nor helps.
They find that retrieval-based models that take advantage of relevant news articles outperform the models without retrieval and become even more effective as the number of parameters increases. However, all these language models still perform significantly worse than the aggregate human forecast (which is a time series of forecasts from start to resolution date).

---

> ### Author Response · Authors · 2022-08-16
> **Response to Reviewer RYpB**
>
> Thank you for your careful analysis of our work. We hope the following response addresses your concerns.
>
> **Metrics.** We have updated the paper with text specifying the directionality of the metrics. Thank you for your suggestion. The combined score metric uses 2 in its denominator so that the maximum score is 100. This is obtained when T/F and MCQ (percent accuracy) are both 100, and Numerical (distance) is 0. Thus, we divide by 2 and not 3.
>
> **Common Crawl News Usage.** The news articles in our Common Crawl news corpus are used for retrieval and are available to the model at training and test time. Using them at test time is not a problem, because for a given question we only use news articles from before the resolution date, so information from the future is never available to the model.
>
> **FiD Static and FiD Temporal.** In FiD Static, we use the top 10 articles from the entire question range. In FiD Temporal, we use the top 1 article from each day during the question range, so FiD Temporal often sees far more articles than FiD Static. These methods are both baselines that we introduce, and they are simply intended to explore different ways of approaching the forecasting task: While FiD Static is a more standard retrieval-based predictor, we use FiD Temporal to demonstrate how the temporal annotations can be used to enable making predictions across time as more news becomes available (see Figure 6 for an example of this).
>
> **Updates to the GitHub.** We have released UnifiedQA code in an updated version of the GitHub repository. The Common Crawl news corpus is now linked to in the GitHub page.
>
> **Other Points.** When we say 512 tokens, we refer to tokens produced by the standard T5 tokenizer (SentencePiece). These are pieces of words.
>
> In Table 2, the Numerical metric is a distance metric, so lower is better. We have updated the paper to clarify this.
>
> The blue line in Figure 1 shows the predictions from a single FiD Temporal model. We have updated the paper to clarify this.
>
> There are no open-ended questions in the dataset. All questions are true/false, multiple choice, or numerical. This is because the forecasting tournaments that we obtain questions from do not support open-ended questions.
>
> The metadata referred to in line 138 includes detailed background information about the question (including precise terms of resolution) and links to relevant information posted by tournament participants. Using this metadata is optional, and for simplicity we do not use it in our baselines, but it could be used to improve performance in future work.
>
> We agree with your point that humans can train on past data, e.g., by studying history. That said, there is truth to the phrase “hindsight is 20/20”, and being able to ‘forget the future’ would surely help one develop stronger forecasting skills and better calibration. That is, while humans can learn from history books, we cannot actually simulate the conditions of true forecasting for past events. We have updated the paper to clarify this point. Thank you for bringing up this important distinction.

---

### Official Review · Reviewer_r5rE · 2022-07-26
**Strong dataset contribution for work on forecasting models**

**Rating:** 8
**Confidence:** 4
**Clarity:** Excellent. All parts of the main pape…

**Strengths:**

- Important, well documented contribution of a novel dataset with very high potential for real world impact.

- Clear and concise paper that explains why this dataset is useful and provides details about how it was collected, and how similar datasets might be collected in the future.

- The paper is convincing that this is a pretty important domain for research, i.e. it is fairly important to have discussions in the community about benchmarks in the forecasting space.

- Early stage evaluations are also provided which may save researchers working with this dataset substantial time.

**Weaknesses:**

Minor: the main paper is sometimes a bit vague in discussing the specific scraping and manual verification process. Specifically, any additional details that can be provded about how the human curation of data was performed will probably be helpful for this kind of work, given that this (1) a dataset contribution and (2) this is the kind of task where people with different backgrounds may make different curation decisions. As I understand, full details on the questions side of the data about how the *questions* were curated may not be possible, as the platforms hiring moderators to curate questions are private platforms.

That said, the general topic of scraping and verification is briefly discussed several times in the supplement and code is available for replication. The info provided in Section 3 is very likely enough for independent replication.

Additionally, were there major differences in the curation aspects of the main dataset vs. the calibration dataset?

It may be worth noting that the test set only lasts for one year (in slight contrast to the example given of using the collapse of the Soviet Union as a motivating example), just to avoid the small chance that readers may think that examples like Soviet Union example appear prominently in this test set.

Finally, it may help to discuss whether a more complete “backtesting” approach from finance is feasible here. Does the LM training cost make this prohibitively expensive? Should researchers in this space be aiming to work towards comprehensive backtests in the long run?

All these concerns are very minor. Hopefully these are helpful in the strengthening the paper, but I’d argue for accepting the paper even with very minor revisions.

**Additional Feedback:**

Thanks to the authors for this contribution!

**Correctness:**

The described train/test split and baseline evaluation seem reasonable. The modeling choices also seem well justified. Evaluation metrics also make sense. A good amount of details about training are provided, (e.g. training epochs, etc.), which I assumed was mainly for reproducibility as no specific results seem to be dependent on training choices here.

The actual data scraping process also seems correct. As noted above, more details about manual checking process would be helpful (random sample? how many observations were manually inspected checked? etc.)

**Documentation:**

In addition to the minor comments above about scraping/curation, it may be worth alluding — in the main paper text — to the kinds of additional info covered in the supplementary materials. I found the supplement very helpful, as it answered many questions that arose, it seems likely other readers may have a similar experience.

**Ethics:**

No major concerns. The attached datasheet is helpful in addressing major questions that might arise, thanks to the authors for providing this.

**Relation To Prior Work:**

Good coverage of recent work in information retrieval, which I think gives reader necessary context to understand how the dataset contribution adds to existing work. Overall, as I understand, this dataset is very novel.

**Summary And Contributions:**

In this paper, the authors describe their efforts to (1) curate a dataset of forecasting questions (taken from forecasting tournaments) and answers (drawn from a news corpus) and (2) provide an early evaluation of popular modeling for at answering these forecasting questions. Great care was taken so that there is no time-based data leakage (i.e., machines can simulate learning facts about the world in the order they occurred).

The framing of the problem is very convincing, i.e. the paper makes a strong argument that this dataset will be important for a variety of applications and fields. The Introduction does a great job motivating the work and summarizing the key contributions. Related Work is focused on situating related technical work on forecasting, IR, QA.

As noted below, the clarity of the work is also great, which enhances the overall contribution.

---

> ### Author Response · Authors · 2022-08-16
> **Response to Review r5rE**
>
> Thank you for your careful analysis of our work. We hope the following response addresses your concerns.
>
> **Curation process.** As you mentioned about the curation process, the forecasting sites we scraped from have comprehensive review processes before a question is posted to their platforms, so questions were of high quality. Although the exact details of the review process may not be accessible, we think the rules and steps outlined here (https://www.metaculus.com/help/faq/#whatsort) offer considerable insights into the curation process.
>
> **Scraping and manual verification process.** Our curation process for Autocast included scraping question texts, compiling human forecasts, and negating the T/F questions with a GPT-3 Edit model. Since the questions were of high quality, and the raw time series data for human forecasts were directly obtained from the websites, we identify question negation to be the most brittle step in the curation process. To address this, all T/F question-negation pairs were manually inspected and revised for correctness and naturalness.
>
> **Curation process for IntervalQA.** The curation process for IntervalQA was fully automated. We collected the questions from existing high-quality datasets such as SQuAD, GSM8K, and MMLU, by filtering for questions with numeric labels.
>
> **Connection to backtesting.** We agree with you that our evaluation is essentially backtesting and a more standardized backtesting framework/library for machine forecasting could improve ease of use and transparency. Currently, this would require substantial engineering effort, but it is certainly an exciting possibility. Nonetheless, we released our source code on GitHub which includes backtesting functionalities, adaptable for other researchers.
>
> **Other Points.** We have updated the paper to make the main paper better incorporate information in the appendix. We also highlight the time period of the test set and show sample test questions in the paper. Thank you for your suggestions.

---

> > ### Comment · Reviewer_r5rE · 2022-08-18
> > **Thanks for this response!**
> >
> > This response is very helpful in addressing the minor questions/concerns raised in my original review, and overall these updates help strengthen the submission.

---

### Official Review · Reviewer_yQuF · 2022-07-27

**Rating:** 6
**Confidence:** 3
**Correctness:** 1. About unresolved questions. Will t…

**Strengths:**

This paper introduces a new dataset based on public forecasting tournaments. It expands the existing future events forecasting QA research. It will bring new research material to the related communities.


**Weaknesses:**

1. About human performance comparison. Based on my understanding, the human aggregated performance is provided by domain experts or enthusiasts (crawled from forecasting tournaments?). Different categories probably have different experts with lots of domain knowledge. However, for a deep learning model, it is very difficult to gain knowledge of all domains. So, the large gap between models and human performance reported in the table is no surprise. From my point of view, a more reasonable human performance collection at the current stage is collecting the performance from multiple crowdsourcing workers and each worker should answer questions from multiple categories (at least one question from one category) using news content before the closing date as input information. I agree that this extra human performance collection would take much extra effort. But I think it would also be a more practical indicator for evaluating further Autocast models.
2. It seems that no information about the distribution of the question types (i.e., T/F, MCQ, and Numerical) in Autocast. It could further strengthen the selection of the combined Score metric. For example, if the majority of the dataset is MCQ as well as T/F questions, a model with a strong ability in these types but worse Numerical performance would also have a relatively high combined score.
3. Why not report the error bars of the investigated models? The results would be more convincing if different random seeds are applied for running experiments multiple times.
4. For UnifiedQA. According to the paper, the reported result is the zero-shot performance (without any fine-tuning?). Is there a particular reason for not fine-tuning UnifiedQA on Autocast but T5 is fine-tuned?

**Additional Feedback:**

1. Please add the venue information for ForecastQA reference in the bib (Line 373). It is accepted here: https://aclanthology.org/2021.acl-long.357/
2. Since the proposed dataset is based on public forecasting tournaments, it would be better to introduce the licensing information of the tournaments (if available).
3. Line 147 states that if a model is pre-trained on more recent data, it will not simulate forecasting faithfully. Any possible measures to avoid this? Otherwise, I think this would affect the future usage of the proposed dataset. For example, future researchers are highly likely to design DL models based on pre-trained language models for Autocast and they probably won't pay much attention to the data used for pre-training (e.g., using off-the-shelf models). It would lead to an unfair benchmark comparison.

**Clarity:**

Overall, the paper is well written.
Please consider reorganising the paper to make the corresponding tables/figures closer to the paragraphs that discuss the tables/figures.

**Documentation:**

The details of the dataset including organization, documentation, and hosting are sufficient. However, the provided codes can only support the reproducibility of FiD methods in the benchmark (no codes for reproducing T5 and UnifiedQA).

**Ethics:**

There is no personally identifiable information and crowd predictions are anonymised/aggregated.
Two potential concerns:
1. Since the data was collected from public forums, were the forum users asked to consent to use the data they produced, and whether the authors have contacted the forecasting tournament organizers' permission to use or share data? Although the Supp file points out that Fair Use §107 is abided, I think it would be better if the authors could state whether they have tried to contact tournaments/forums.
2. When collecting the dataset from public forums and news articles, it is possible that the forums contain offensive content. The description of how are the authors filtering offensive content such as racist language or violent imagery to guarantee that these contents are absolutely avoided in the dataset is not given in the submitted files.

**Relation To Prior Work:**

1. Missing prior related work in the discussion:
[1] A Dataset for Answering Time-Sensitive Questions (neurips21, dataset track)
[2] Improving Time Sensitivity for Question Answering over Temporal Knowledge Graphs (ACL)
Especially, I think TimeQA (reference [1]) is a relevant paper that should be considered.

2. Since this work is very close to the previous ForecastQA, I suggest including a statistical comparison (e.g., number of data instances, date ranges, etc.) of Autocast and ForecastQA.

**Summary And Contributions:**

The authors propose a new Autocast dataset for future events forecasting under the QA format. The raw data comes from public forecasting tournaments, which ensures the quality of the event forecasting questions. The dataset also includes news corpora corresponding to each question that can be leveraged by forecasting models. A benchmark including human performance is also introduced.

---

> ### Author Response · Authors · 2022-08-16
> **Response to Reviewer yQuF (1/2)**
>
> Reviewer yQuF,
>
> Thank you for your careful analysis of our work. We hope the following response addresses your concerns.
>
> **Human Performance Comparison.** Human crowd performance does tend to exceed that of individual humans, which you correctly note is due to wisdom of the crowd and self-selection based on expertise. However, we think human crowd performance is actually very reasonable to compare to. This is because (1) large language models pre-trained on the entire internet acquire extensive world knowledge (https://arxiv.org/abs/2009.03300), and thus have a strong intrinsic advantage over individual human forecasters; and (2) crowd predictions are the median predictions of individual forecasters, so the individual forecasters cannot be substantially worse (ensembling only goes so far). It would be interesting to see how MTurkers compare, but you are right that this data would be challenging to collect, because they would have unrealistic access to information from after the resolution dates of questions.
>
> **Distribution of Question Types.** We have added this information to the appendix in the updated paper. Thank you for your suggestion.
>
> **Error Bars.** Due to the high computational cost of fine-tuning large language models, running all of our experiments takes approximately 2 weeks, so we do not compute error bars. Also note that there is much room for improvement over the baselines, so we do not expect noise due to random initializations to mask improvements.
>
> **UnifiedQA.** UnifiedQA is intended to be our zero-shot baseline. This is because UnifiedQA is already fine-tuned on a wide variety of QA formats, allowing it to be used in a zero-shot manner on QA tasks. By contrast, T5 requires fine-tuning. In fact, zero-shot UnifiedQA can substantially outperform GPT-3 on QA tasks (see Table 1 in https://arxiv.org/abs/2009.03300). However, it barely improves over random performance on Autocast, indicating that strong performance at forecasting may require new methods (i.e., simply scaling up is not enough).
>
> **Unresolved Questions.** For future Autocast versions, previously unresolved questions will be updated if they have resolved (i.e., we will replace the unresolved questions with their resolved counterparts). We will also update unresolved questions to include the latest crowd forecasts. For our baselines, we started off with only using resolved questions, which simplifies the data loader and training loss, but we do hope to experiment with incorporating unresolved questions in future work. We suspect this could improve performance, especially since there is a large gap between the crowd and model performance, which means that human forecasts are a valuable training signal, albeit not quite as good as ground-truth resolutions.
>
> **Prior Work.** Thank you for pointing us to these works. TimeQA is especially relevant, and we have updated the paper to include a citation. We have also fixed the ForecastQA citation.
> Due to the differences between ForecastQA and Autocast in the curation process, many dataset statistics are hard to compare. For example, each question in Autocast has a date range during which people can make forecasts but there’s no such notion in ForecastQA. If we consider the date range during which the events occured, ForecastQA only spans 2019 from which the gold articles are collected, whereas Autocast spans a much wider range - from 2016 to the present for resolved questions and far into the future for unresolved questions. Currently, the two datasets contain nearly the same number of questions (roughly 10,000), but Autocast dynamically grows over time whereas ForecastQA doesn’t. For the above differences and potential confusions, we were sufficiently disincentivized to include such comparisons.
>
> **UnifiedQA Code.** We have released UnifiedQA code in an updated version of the GitHub repository.

---

> ### Author Response · Authors · 2022-08-16
> **Response to Reviewer yQuF (2/2)**
>
> **Permission for Data Usage.** Since submitting the paper for review, we have obtained full permission from Metaculus for using their forecasting questions. The majority of questions in Autocast come from Metaculus, and for the remaining questions we abide by Fair Use §107. Due to the nature of online data collection, it is extremely challenging to seek consent from individual authors of forecast predictions. However, as we note in the Data Sheet in our Appendix, information from individual users is aggregated into community forecasts and thus does not include any personally identifiable information.
>
> **Content Moderation.** All questions in Autocast are rigorously reviewed from an ethics perspective according to the moderation policies of the original forecasting tournaments. For Metaculus, the moderation rules can be found here: https://www.metaculus.com/question-writing/ and include “Questions should not contain inappropriate or offensive material”, “Public questions should not concern the personal lives of non-public figures”, etc. For Good Judgment, the moderation rules can be found here: https://www.gjopen.com/faq and include that all content (including questions) cannot be “harmful, threatening, unlawful, defamatory, infringing, …” Content moderation policies for the forecasting tournaments apply to forum posts/comments as well, but we do not include text from forum posts, so this is not a concern. We have updated the paper to include this information thanks to your suggestion.
>
> **Faithfully Simulating Forecasting.** You are correct that models used for Autocast should not be pretrained on data that was created after the test split date of 5-11-2021. Fortunately, many models do meet this requirement, including BERT, XLNet, T5-v1.1, DeBERTa models, UnifiedQA models, GPT-2 models, GPT-3 (001 models), GPT-Neo models, LaMDA, Gopher, Chinchilla, or models that trained only on the pre-training data used by the aforementioned models. Thus, we do not expect this to be a significant constraint on future work. In future versions of Autocast, we may also push the test set date forward to allow for newer models if needed.

---

> ### Comment · Reviewer_yQuF · 2022-08-26
> **Response acknowledged**
>
>
> Thanks for the efforts made by the authors.
> I think my question is well answered to some extent. I would like to keep my ratings.

---

### Official Review · Reviewer_58NQ · 2022-07-27
**Interesting paper**

**Rating:** 6
**Confidence:** 3
**Clarity:** yes

**Strengths:**

- The paper is interesting and well-written.
- The dataset has **high-quality** forecasting questions and it has an associated **temporal** news corpus. The problem is somehow well-defined.
- It design some simple baselines and the results seem making-sense.

**Weaknesses:**

- I am not sure if all these questions are **predictable** given the *temporal* news corpus (especially top ten news based on a ranker). Could  the top-ranked news  provide enough inductive bias to predict the questions？This might not make sense from agnosticism point of view.
- FiD Temporal does not (significantly) outperform FiD Static. One might wonder what does this really mean for  incorporating the auxiliary crowd predictions.
- In line 235, should Table 3 be Table **2**?

**Additional Feedback:**

see weakness above

**Correctness:**

it is constructed in a sound way. But the soundness should be further discussed, for exmaple, should all these questions be predictable from these temporal news?

**Documentation:**

good

**Relation To Prior Work:**

seems good

**Summary And Contributions:**

This paper introduces Autocast dataset that contains thousands of  **high-quality** forecasting questions and an accompanying  **temporal** news corpus. The paper is interesting and well-written.

---

> ### Author Response · Authors · 2022-08-16
> **Response to Reviewer 58NQ**
>
> Thank you for your careful analysis of our work. We hope the following response addresses your concerns.
>
> > I am not sure if all these questions are predictable given the temporal news corpus (especially top ten news based on a ranker). Could the top-ranked news provide enough inductive bias to predict the questions?
>
> This is a good question. Our Common Crawl news corpus contains 200GB of text and essentially covers all news items in the world between 2016 and 2022. Most questions in Autocast have multiple relevant news articles in the corpus (see Fig. 5 in the paper for a representative example). However, some questions may not be easily answerable without additional sources of information, such as research papers, reports, and multimodal data. For this reason, we do not require future work to only use the news corpus from this paper. As long as all data used for answering a question predates the resolution date, using additional data is fair game. In fact, we think this is an exciting direction for future work. We will clarify this in the updated paper.
>
> > FiD Temporal does not (significantly) outperform FiD Static.
>
> The FiD Temporal method is primarily for demonstrating one possible way of using the temporal crowd predictions as auxiliary supervision, which facilitates analyzing how model predictions change over time (this is not possible with the FiD Static model). We hope that future work will be able to make better use of the crowd predictions, which contain rich temporal information for resolved and unresolved questions.
>
> > In line 235, should Table 3 be Table 2?
>
> Yes, it should. Thank you for pointing out this typo. We have fixed it in the updated paper.

---

> > ### Comment · Reviewer_58NQ · 2022-08-24
> > **Thanks for this response!**
> >
> > Thanks for this response! I would like see it accepted in this venue.

---

### Official Review · Reviewer_qpTi · 2022-07-28
**Brief review of "Forecasting Future World Events With Neural Networks" paper.**

**Rating:** 7
**Confidence:** 3
**Clarity:** The paper is well written and easy to…

**Strengths:**

Well structured and mostly well described, easy to read, the difference from ForecastQA is significant, so the contribution is considerable.

**Weaknesses:**

It is not clear how to use "source_links" field and how CC news corpus is used.

Some article URLs from the dataset are unavailable: some redirect to another page, some require subscription.


**Additional Feedback:**

I would add a more detailed example from the dataset, i.e. how the crowd decision distribution changed over time, which news caused that and why.

Also the instruction for "source_links" field processing would be great.

**Correctness:**

Overall dataset structure is fine, but it is not clear for me how to retrieve articles from provided links correcty.

Evaluation methods and experiment design looks correct.

**Documentation:**

The provided Github link is working and the dataset is available, everything seems to be fine. There is an ipynb script showing how to work with this dataset.

**Ethics:**

The dataset consists of various questions and possible answers from public tournaments. Human participation here is answering these questions using different sources of information, I don't see any ethical conserns in this action. The only possible problem is: are these questions already reviewed from ethics perspective?

**Relation To Prior Work:**

It is described clearly how AutoCast dataset differs from ForecastQA dataset.

**Summary And Contributions:**

The Autocast dataset was introduced, a dataset for measuring the ability of neural networks to forecast future world events. Autocast poses a novel challenge for large language models and improved performance could bring large practical benefits. Additionally, Autocast questions were written by experienced forecasters and are always unambiguous given the full question description, in contrast to related work.

---

> ### Author Response · Authors · 2022-08-16
> **Response to Reviewer qpTi**
>
> Thank you for your careful analysis of our work. We hope the following response addresses your concerns.
>
> > It is not clear how to use "source_links" field and how CC news corpus is used. Some article URLs from the dataset are unavailable: some redirect to another page, some require subscription.
>
> Thank you for bringing this to our attention. The source_links field contains links that were referenced in the human forecasters’ comments. There is not much we can do about paywalls and redirects in the posted links, but we will add a disclaimer in the paper that this information may be less reliable than other annotations due to these general challenges. These links are meant to provide additional signals and can be used in a variety of ways if one chooses to. For example, one can imagine training a better retriever because these links can be considered gold articles to the questions. Or one can imagine directly training the forecaster model by incorporating the articles from these links. However, we leave this exploration for future work and only retrieve from the CC-NEWS corpus in our paper. More specifically, given a question, for each day the question is active, we retrieve the top 10 relevant news articles from the daily articles. In our FiD-Temporal experiments, we only use the top 1 from every day. Then, we aggregate all these articles from different dates and rank them according to the retrieval score. The top 10 articles are used for the FiD-Static model. We have updated the paper to include this information thanks to your suggestion.
>
> > The only possible problem is: are these questions already reviewed from ethics perspective?
>
> All questions in Autocast are rigorously reviewed from an ethics perspective according to the moderation policies of the original forecasting tournaments. For Metaculus, the moderation rules can be found here: https://www.metaculus.com/question-writing/ and include “Questions should not contain inappropriate or offensive material”, “Public questions should not concern the personal lives of non-public figures”, etc. For Good Judgment, the moderation rules can be found here: https://www.gjopen.com/faq and include that all content (including questions) cannot be “harmful, threatening, unlawful, defamatory, infringing, …”
>
> > I would add a more detailed example from the dataset, i.e. how the crowd decision distribution changed over time, which news caused that and why.
>
> We agree that this would be interesting to add. We have added an expanded version of Figure 1 to the appendix, showing how the crowd forecast is influenced by news articles posted to the question page. There are clearly articles that have a large impact, but not all articles affect the forecast, and the forecast tends to change even when articles have not been posted. This is because new forecasters join the question at various times, and forecasters check on the question and make predictions asynchronously, which tends to result in a smoothly varying crowd forecast over time.

---

> > ### Comment · Reviewer_qpTi · 2022-08-29
> > **Thanks for your response**
> >
> > Thanks for your response! My questions are well answered.

---

### Review · Ethics_Reviewer_nK2v · 2022-08-26

**Recommendation:** 2

**Ethics Documentation:**

The authors' reliance on 17 USC 107 is legally and facially insufficient to overcome the Terms of Use (TOU) as plainly provided on the Metaculus website, as an example. The relevant TOU are reproduced in detail in quotations above for ease of reference for the authors.

Accordingly, the authors need written permission of the Metaculus and Common Crawl providers, as examples, to scrape and use their site content.

Their express terms of service also hold third-parties accountable, which would include this Conference.

Therefore a predicate to moving forward with the paper submission is the requirement for written approval of the sites scraped according to the Terms of Use of each dataset scraped.

If such written approval or evidence of compliance cannot be obtained, then it is suggested that the model be recalibrated without that unauthorized dataset.
Otherwise, the datasheet should be updated to note that the terms of use have been complied with. The authors may wish to seek legal guidance from their sponsoring or employing entity, whether a university, company, or governmental entity as third-party affiliation may also extend to those entities.

It is ethically advisable to ask for permission rather than forgiveness. The consequences for failing to ask may be quite serious.

**Ethics Review:**

The Ethical Review Guidelines used in this ethics review may be found at https://neurips.cc/public/EthicsGuidelines.

The authors created the Autocast dataset to evaluate state-of-the-art neural networks on the task of judgmental forecasting.
Data were also collected from the Common Crawl news corpus and IntervalQA, a large collection of numerical prediction questions with a wide dynamic range of prediction targets. Large language models (LLMs) were used for the assessment.
There is no personally identifiable information in the data and crowd predictions are anonymised/aggregated.

However, the Autocast dataset was collected by scraping questions from public forecasting platforms on the internet: Metaculus, Good Judgment, and CSET-Foretell, that contained thousands of forecasting questions from public forecasting tournaments, including ground truth outcomes and aggregated human predictions.

The authors' use of the Datasheet Q&A format answered many of the ethical concerns/questions and showed that the authors had a fundamental understanding and appreciation for the ethical concerns encountered in their work and generation of the dataset.

No consent was obtained directly by the authors because the data were collected from public forums where the forum users obtained consent for the data. However, there is evidence of copyright violations and violations of Terms of Use (TOU) of the scraped websites from which the data were collected.

The authors rely on fair use to avoid copyright infringement, citing 17 USC 107.
This reliance is insufficient and misplaced in light of the express language in the terms of use of at least the Metaculus website.
The authors should be aware that the matter of copyright and fair use is not legally settled in the US with regard to data scraping and machine learning. Data transformation in a machine learning algorithm is not a legal consideration under US Copyright law.
Additionally, the GDPR and PIPL have different standards.
It is strongly suggested that the authors review Mark Lemley and Bryan Casey's Law Review Article entitled Fair Learning, published in the Texas Law Review, Vol 99, Issue 4 (March 2021) at https://texaslawreview.org/fair-learning/, which provides a fairly recent update on the law of fair use with particular focus on machine learning. The article includes discussions of various jurisdictions around the world and notes their differences. Not all areas have the same interpretation of fair use and the limits and applications of fair use as applied to data and machine learning are discussed at length in various contexts.

As an example, the review looked at the Metaculus website. It has a Terms of Use on their website at https://www.metaculus.com/terms-of-use/
The TOU expressly provides limitation to the user, which includes the following language:
"You agree not to view, copy, or procure content or information from the Service by automated means (such as scripts, bots, spiders, crawlers, or scrapers), or to use other data mining technology or processes to frame, mask, extract data or other materials from the Metaculus Content (except as may be a result of standard search engine or Internet browser usage), unless formally authorized by Metaculus under separate written agreement. No materials from the Service may be copied, reproduced, modified, republished, downloaded, uploaded, posted, transmitted, or distributed in any form or by any means without Metaculus's prior written permission or as expressly provided in these Terms of Use. When you download or use the Metaculus Content as authorized by these Terms of Use, you must: (a) keep intact all copyright and other proprietary notices; (b) make no modifications to the Metaculus Content; and (c) not copy or adapt any object code associated with the Service or reverse engineer, modify or attempt to discover any source code associated with the Service, nor allow or assist any third party (whether or not for your benefit) to do so. All rights not expressly granted herein are reserved. Metaculus may impose reasonable limits on your scope of access to Metaculus Content, including limits on time or number of materials accessed or machines used to access such Content, to prevent unauthorized third party access to or use of that Content."

The authors' reliance on 17 USC 107 is legally and facially insufficient to overcome the TOU as plainly provided on the Metaculus website.
Accordingly, the authors need written permission of the Metaculus providers to scrape and use their content.
Their express terms of service also hold third-parties accountable, which would include this conference.

There are similar issues with the Terms of Use for the Common Crawl website https://commoncrawl.org/terms-of-use/full/.

Therefore a predicate to moving forward with the paper submission is the requirement for written approval of EACH of the sites scraped according to their Terms of Use.

If such written approval or evidence of compliance cannot be obtained, then it is suggested that the model be recalibrated without that unauthorized dataset.

The reviewer appreciates that data are sometimes difficult to obtain. The reviewer is also experienced with once-open-source scrapable databases transitioning to fee-for-service systems. There are instances where some of these sites may consent to scraping or scraping for educational or research purposes. If no overt Terms of Use are set forth permitting scraping, it is advisable to request a written notice from the database administrators or whomever controls the content authorizing the use of the scraped data expressly for educational or research purposes. If the site declines the authorization, the data may not be used because it has been unethically obtained and potentially obtained in violation of applicable law.

---

> ### Comment · Area_Chair_87tr · 2022-08-28
> **Authors, please address the ethics review**
>
> Dear authors, we need you to address the ethics review and follow the recommendations carefully. Please submit the revision along with a written approval from each of the scraped websites by the deadline, otherwise, you may want to consider a mitigation, i.e. recalibrated model without the authorised dataset.
> Looking forward to reading your response on the ethics review.

---

> ### Author Response · Authors · 2022-08-31
> **Response to Ethics Reviewer nK2v**
>
> Thank you for your careful analysis of our work. With the help of your suggestions, we have resolved clarity issues regarding the ethical questions you raise. We apologize for not making our legal compliance more clear in the original submission (e.g., we received full permission months ago from Metaculus’s CEO, Gaia Dempsey), and we hope the following response addresses your concerns.
>
> **We Received Full Permission Months Ago**
>
> We agree that the Metaculus terms of use are quite stringent, such that Fair Use protections are not sufficient. For that reason, we reached out to Metaculus’ CEO, Gaia Dempsey, and obtained consent for collecting data for Autocast months ago (see here: https://drive.google.com/file/d/12fSsLsmnODr2obA2c7kuY921Thgrv3eo/view?usp=sharing). It is also worth mentioning that Metaculus has even approved the community to make forecasts about our dataset (https://www.metaculus.com/questions/11670/when-will-ml-make-near-human-level-forecasts/).
> The approval from Gaia has been reflected in the statement on our GitHub indicating that they have granted us permission. Although it was emphasized on GitHub, we apologize that we did not also emphasize this inside the appendix, which has resulted in this confusion. We have updated the paper to clearly reflect our compliance. If Metaculus were to rescind permission in the future, we would be happy to amend the dataset and train recalibrated models.
>
> **Permission from Other Websites**
>
> CSET is a policy research organization within Georgetown University, and Foretell was a “crowd forecasting pilot project focused on technology and security policy.” The CSET-Foretell website does not have a Terms of Use page, and we only use the data for academic, non-commercial purposes, so Fair Use protections are sufficient. Similarly Good Judgment, which was created by professors and researchers at Penn, relies on the same forecasting platform as CSET, and we reached out and had a zoom conversation with Good Judgment executives early on in the dataset curation process. Please note that one of our authors, Jacob Steinhardt, has worked with CSET employees previously, and CSET employees are generally supportive of this paper and forecasting work. (As it happens, Jacob has also co-written a paper for the Brookings Institution with the main director of CSET, Helen Toner.) Since creating a formal contract would be a slow, costly, and nearly unprecedented requirement for academic machine learning research, we are proceeding under Fair Use protections for the time being, as these are sufficient. (We agree that for Metaculus more permission was necessary, which we obtained as well.) Going forward, we also intend to modify the dataset across time to be in keeping with copyright law. For instance, we may add questions from the Manifold Markets (https://manifold.markets) forecasting platform to a future version of Autocast, and for these questions we already have permission from the Manifold Markets CEO, Austin Chen, who also assisted us in efficiently curating the data from the platform. Consequently, we have been and will continue to be proactive about copyright compliance.
>
> **Legal Compliance**
>
> > “Their express terms of service also hold third-parties accountable, which would include this conference.”
>
> As we note in Appendix B, we bear all responsibility in case of violation of rights. Additionally, the language regarding third parties pointed out by the reviewer is only the case in the TOU for Metaculus, for which we have received full permission for usage. We would like to stress to the AC in particular that there is *very little* potential for liability, especially considering the non-commercial, academic nature of the work and the existence of Fair Use protections.
>
> **Common Crawl News**
>
> Please note that the Common Crawl terms of use permits usage of the data for non-harmful use cases (https://commoncrawl.org/terms-of-use/full/). Thus, we follow the Terms of Use for the Common Crawl website. However, to take extra precautions, we have removed the link to our cc_news dataset from our Github page. We make the dataset fully reproducible by including the script to download and filter cc_news on GitHub.
>
> We have updated the appendix and datasheet to reflect all changes. We have also improved clarity for how we are fully legally compliant and have been proactive about copyright compliance. Thank you very much for your suggestions.

---

### Meta-Review · Area_Chair_87tr · 2022-09-09

**Recommendation:** Accept
**Confidence:** 4

**Metareview:**

The paper presents a novel dataset on the task of judgmental forecasting. Autocast was collected by scraping questions from public forecasting tournaments, accompanied by a news corpora corresponding to each question. The reviewers agree that this is a high quality and novel dataset with diverse questions and tasks. It has a potential for high impact given the benefit of the datasets. The paper is clearly written, accompanied by a detailed data sheet.
After consulting the ethics chairs, I would like to recommend conditional acceptance of this paper, pending minor revision.

The authors clarify that they received permission from the source websites when possible, directly addressing the concerns of Ethics Reviewer nK2v and Reviewer qpTi.  This additional context needs to be mentioned explicitly in the paper and supplementary materials.
Authors should state the following in an appendix or footnote to the main text:
Full permission for Metaculus CEO, Gaia Dempsey
CSET-Foretell website does not have a Terms of Use but the data is only being used for academic, non-commercial purpose, which is consistent with CSET’s purpose as a policy research organization and fair use.
The Terms of Use for the Common Crawl website are being used. However, to further compliance, the link to the cc_news dataset is removed from our GitHub page. The dataset is fully reproducible by including the script to download and filter cc_news on GitHub.

The terms of use of the involved platforms addresses the concern for the presence of potentially offensive content.

---

### Decision · Program_Chairs · 2022-09-16

Accept